# DM-Codec: Distilling Multimodal Representations for Speech Tokenization

## Abstract

Recent advancements in speech-language models have yielded significant improvements in speech tokenization and synthesis. However, effectively mapping the complex, multidimensional attributes of speech into discrete tokens remains challenging. This process demands acoustic, semantic, and contextual information for precise speech representations. Existing speech representations generally fall into two categories: acoustic tokens from audio codecs and semantic tokens from speech self-supervised learning models. Although recent efforts have unified acoustic and semantic tokens for improved performance, they overlook the crucial role of contextual representation in comprehensive speech modeling. Our empirical investigations reveal that the absence of contextual representations results in elevated Word Error Rate (WER) and Word Information Lost (WIL) scores in speech transcriptions. To address these limitations, we propose two novel distillation approaches: (1) a language model (LM)-guided distillation method that incorporates contextual information, and (2) a combined LM and self-supervised speech model (SM)-guided distillation technique that effectively distills multimodal representations (acoustic, semantic, and contextual) into a comprehensive speech tokenizer, termed DM-Codec. The DM-Codec architecture adopts a streamlined encoder-decoder framework with a Residual Vector Quantizer (RVQ) and incorporates the LM and SM during the training process. Experiments show DM-Codec significantly outperforms state-of-the-art speech tokenization models, reducing WER by up to 13.46%, WIL by 9.82%, and improving speech quality by 5.84% and intelligibility by 1.85% on the LibriSpeech benchmark dataset.

## 1 Introduction

In recent years, the advent of Large Language Models (LLMs) has revolutionized various domains, offering unprecedented advancements across a wide array of tasks (OpenAI, 2024). A critical component of this success has been the tokenization of input data, enabling vast amounts of information processing (Du et al., 2024; Rust et al., 2021). Inspired by these breakthroughs, significant attention has shifted towards replicating similar successes in the realm of speech understanding and generation (Défossez et al., 2022; Hsu et al., 2021). However, tokenizing speech into discrete units presents unique challenges compared to text, as speech is inherently continuous and multidimensional, requiring various speech attributes such as acoustic properties, semantic meaning, and contextual clues (Ju et al., 2024). Traditional approaches using feature representations such as Mel-Spectrograms (Sheng et al., 2019), Mel-frequency cepstral coefficients (MFCCs) (Juvela et al., 2018), and Waveforms (Kim et al., 2021) have proven inadequate in capturing this full spectrum of information, resulting in suboptimal performance in downstream tasks such as speech synthesis (Ju et al., 2024).

These limitations led researchers to explore various approaches, and one prominent direction leading to audio codecs (Borsos et al., 2023). Notable examples include SoundStream (Zeghidour et al., 2021) and EnCodec (Défossez et al., 2022), which utilize Residual Vector Quantizers (RVQ) within a neural codec framework, iteratively refining quantized vectors to discretize speech into acoustic tokens. Concurrently, self-supervised speech representation learning models such as HuBERT (Hsu et al., 2021) and wav2vec 2.0 (Baevski et al., 2020) facilitated extracting speech representations as semantic tokens (Borsos et al., 2023). Efforts to unify acoustic and semantic representations have led to two notable approaches: SpeechTokenizer (Zhang et al., 2024a), which utilizes semantic dis-

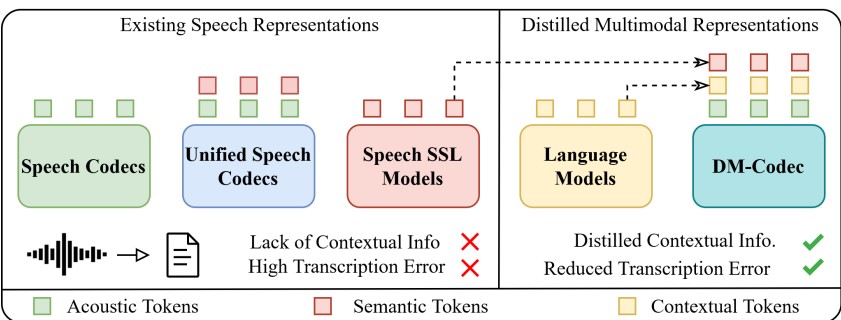

Figure 1: Overview of speech tokenization using discrete acoustic, semantic, and contextual tokens. DM-Codec integrates these features for robust and comprehensive speech representation.

tillation from HuBERT, and FACodec (Ju et al., 2024), which proposes a factorized vector quantizer to disentangle speech representation into different subspaces using separate RVQs with supervision.

While these approaches have shown promising results, they often overlook a crucial aspect of speech representation: the integration of contextual language information. Language models (LMs) have demonstrated a remarkable ability to learn contextual representations that capture the meaning of tokens based on their broader linguistic context (Devlin et al., 2019). These contextual representations can provide essential insights into speech representation, allowing for a more nuanced understanding of words in varying linguistic contexts. Our empirical investigations also reveal that existing discrete speech representation models struggle to align reconstructed speech with accurate textual form, resulting in elevated Word Error Rates (WER) and Word Information Lost (WIL) scores in speech transcription tasks. This observation underscores the need for a more comprehensive approach to speech tokenization that incorporates contextual language information.

To address these challenges, we propose DM-Codec, a novel speech tokenizer that unifies multimodal language and speech representations. Our approach builds on a neural codec architecture incorporating RVQ with encoder, decoder, and discriminator components. Central to our innovation is the introduction of an LM-guided distillation method that effectively incorporates contextual representations into the speech tokenization process. This technique allows DM-Codec capturing the nuances of linguistic context often missed by existing models. Building upon the LM-guided approach, we further propose a hybrid distillation method combining both LM and speech model (SM)-guided techniques. Moreover, we introduce a [CLS]-token-based distillation strategy that leverages sequence-level holistic representations from the LM, effectively capturing global contextual information. To the best of our knowledge, we are the first to attempt to integrate all three essential aspects of speech representation—acoustic, semantic, and contextual—within a single codec. See Figure 1 for a depiction. In addition, to demonstrate the impact of multimodal representation and generalizability of DM-Codec in downstream tasks, we introduce DM-Codec-TTS, a novel multimodal representation distilled neural codec-based text-to-speech model.

Through extensive experimentation on the LibriSpeech benchmark dataset (Panayotov et al., 2015), we demonstrate the superiority of DM-Codec, which achieves significantly lower WER and WIL compared to state-of-the-art baseline speech tokenizers. Specifically, DM-Codec achieves a WER of 4.05 and a WIL of 6.61, outperforming SpeechTokenizer (4.49, 7.10), FACodec (4.68, 7.33), and EnCodec (4.53, 7.17). Furthermore, DM-Codec exhibits improved speech quality, as evidenced by its Virtual Speech Quality Objective Listener (ViSQOL) score of 3.26 and human evaluated Mean Opinion Score (MOS) of 3.72, surpassing the performance of EnCodec (3.08, 3.09), SpeechTokenizer (3.09, 3.67), and FACodec(3.13, 3.70). Similarly, DM-Codec-TTS excels in LibriSpeech and VCTK evaluations, outperforming USLM and Vall-E in content preservation and naturalness. On LibriSpeech, it achieves a WER of 5.08, WIL of 7.32, MOS of 3.70, and SMOS of 3.89, while on VCTK, it achieves a WER of 3.58, WIL of 5.65, MOS of 3.78, and SMOS of 3.85. Notably, DM-Codec-TTS-small achieves a WER of 10.26, WIL of 13.79, MOS of 3.24, and SMOS of 3.20, surpassing USLM_libri across all metrics despite using the same smaller dataset.

Our research makes the following key contributions:

- We introduce DM-Codec, a novel speech tokenizer that incorporates contextual representations via the LM-guided distillation method.

- We present a novel combined LM and SM-guided representation distillation approach, uniting acoustic, semantic, and contextual representations into a unified framework.

- We propose a [CLS]-token-based distillation strategy that captures global contextual information from the LM, facilitating better alignment and transfer of contextual features.

- We introduce DM-Codec-TTS, a novel multimodal representation distilled neural codec-based text-to-speech model, demonstrating the applicability and generalizability of the DM-Codec framework in downstream speech synthesis tasks.

- Through comprehensive experiments and ablation studies, we demonstrate the effectiveness of DM-Codec in preserving increased contextual information and enhancing the retention of acoustic and speech information in reconstructed speech

## 2 PROPOSED METHOD

In this section, we present DM-Codec, a novel speech tokenizer designed to encapsulate a comprehensive fusion of multimodal (acoustic, semantic, and contextual) representations. As illustrated in Figure 2, we propose two distinct training approaches to incorporate these representations: (i) a language model (LM)-guided distillation method, and (ii) a combined LM and self-supervised speech model (SM)-guided distillation method. The first approach distills contextual representations from the LM and integrates them with learned acoustic representations. The second approach combines SM and LM to further incorporate semantic representations with contextual and acoustic representations. It ensures that DM-Codec captures the essential elements of speech by harmonizing the acoustic features with contextual and semantic information. In addition, we propose a [CLS]-token-based distillation method, which focuses on leveraging the holistic sequence-level representations encoded by the [CLS] tokens of the LM. Moreover, we introduce the DM-Codec-TTS, a novel multimodal representation distilled neural codec-based text-to-speech model that incorporates DM-Codec into a neural codec language model architecture. The following subsections detail our proposed distillation methods (§2.1), DM-Codec model details (§2.2) and training objectives (§2.3), and DM-Codec-TTS model details and training objectives (§2.4).

### 2.1 SPEECH AND LANGUAGE MODEL GUIDED DISTILLATION

Our approach first transcribes the raw speech $\mathbf{x}$ into its corresponding text $\mathbf{x}'$ using an Automatic Speech Recognition (ASR) model $M_{ASR}$, such that $\mathbf{x}' = M_{ASR}(\mathbf{x})$, serving as an automation tool for converting audio to text, with no other influence on later steps. For simplicity, we omit any post-processing techniques on the $\mathbf{x}'$. Subsequently, we pass the text $\mathbf{x}'$ through a pretrained language model $M_{LM}$ to obtain contextual representations of $\mathbf{x}'$, tokenized into a set of tokens, $\mathcal{T} = \{t_i\}_{i=1}^n$. For each token $t_i$, we extract its corresponding layer-wise hidden representations $\{\mathbf{h}_i^l\}_{l=1}^L$, where $L$ denotes the total number of layers in $M_{LM}$. We utilize all layer representations to derive the representations for each token, as each layer of a pre-trained language model captures hierarchical and contextually distinct information (Niu et al., 2022; Kovaleva et al., 2019; Hao et al., 2019). To obtain the contextual representation $\mathbf{S}_i$ for token $t_i$, we average the hidden representations across all layers, yielding $\mathbf{S}_i = \frac{1}{L} \sum_{l=1}^L \mathbf{h}_i^l$, where $\mathbf{S}_i \in \mathbb{R}^D$ where $D$ is the hidden dimension. Consequently, we obtain the contextual representations $\mathbf{S} = \{\mathbf{S}_i\}_{i=1}^n$ for the speech input $\mathbf{x}$, which captures the contextually diverse information from $M_{LM}$.

Simultaneously, we process the raw speech $\mathbf{x}$ through an Encoder $\mathbf{E}(\mathbf{x})$ to obtain the latent feature $\mathbf{v}$, with sequence length $T'$. We then pass $\mathbf{v}$ through a Residual Vector Quantizer (RVQ) to obtain quantized features $\mathbf{Q} = \{\mathbf{Q}_k\}_{k=1}^K$, where $K$ represents the number of quantization layers in the RVQ, and $\mathbf{Q}_k \in \mathbb{R}^{D'}$ where $D'$ is the hidden dimension of $k^{th}$ RVQ layer. These quantized features are subsequently used to reconstruct the audio $\hat{\mathbf{x}}$ via a decoder. To align the quantized feature $\mathbf{Q}_k$ with the LM distilled features $\mathbf{S}_i$, we apply a linear transformation $\mathbf{Q}_k' = \mathbf{W}\mathbf{Q}_k$, where $\mathbf{W} \in \mathbb{R}^{D' \times D}$, ensuring the dimensional consistency for the distillation process. To match the sequence length of $\mathbf{Q}_k$ with $\mathbf{S}_i$, we pad the tokens $n$ to the latent acoustic sequence length $T'$.

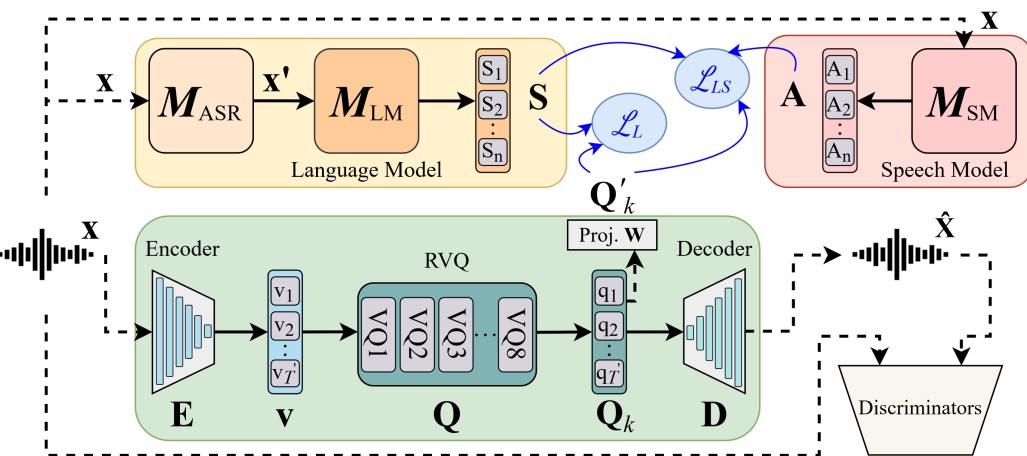

Figure 2: DM-Codec framework consists of an encoder that extracts latent speech representations, quantized using a Residual Vector Quantizer (RVQ). We propose two distillation approaches: (i) from a language model (LM), and (ii) from both an LM and a speech model (SM), integrating acoustic, semantic, and contextual features to enhance speech representations for downstream tasks.

**LM Guided Distillation:** In this approach, we distil the LM representations $\mathbf{S}$. To calculate the distillation loss, we adopt a *continuous representation distillation* technique, similar to the one employed by SpeechTokenizer (Zhang et al., 2024a), which maximizes the cosine similarity at the dimension level across all time steps. In our case, we calculate the continuous representation distillation of the transformed quantized features $\mathbf{Q}'_k$ and the LM representation features $\mathbf{S}$ as follows:

$$\mathcal{L}_L = -\frac{1}{D} \sum_{d=1}^{D} \log \left( \sigma \left( \frac{\mathbf{Q}'^{(:,d)}_k \cdot \mathbf{S}^{(:,d)}}{\|\mathbf{Q}'^{(:,d)}_k\| \|\mathbf{S}^{(:,d)}\|} \right) \right) \tag{1}$$

Here, the notation $(:, d)$ indicates a vector that includes values from all time steps at the $d^{th}$ dimension. The function $\sigma(\cdot)$ represents the sigmoid activation function.

**Combined LM and SM Guided Distillation:** To further enhance the capabilities of DM-Codec, we propose a hybrid approach that utilizes both audio and text modalities. To derive semantic representations from the speech model (SM), we adopt a similar distillation strategy as we used for the LM. We first pass the raw speech $\mathbf{x}$ through the pretrained speech model $M_{SM}$, which generates its own set of layer-wise hidden representations $\{\mathbf{h}^l_j\}_{l=1}^{L}$. The semantic features are derived by averaging the hidden states across all layers, yielding $\mathbf{A}_j = \frac{1}{L} \sum_{l=1}^{L} \mathbf{h}^l_j$, where $\mathbf{A}_j \in \mathbb{R}^D$. This process results in the semantic representations $\mathbf{A} = \{\mathbf{A}_j\}_{j=1}^{n}$ for the speech input $\mathbf{x}$. The distillation loss in this case considers both the LM and SM representations, jointly optimizing for the alignment of the quantized features $\mathbf{Q}'_k$ with the representations $\mathbf{A}$ and $\mathbf{S}$ derived from $M_{SM}$ and $M_{LM}$, respectively. Finally, the distillation loss for the SM, $\mathcal{L}_S$, is first computed, followed by averaging with the LM distillation loss, $\mathcal{L}_L$, to ensure a balanced contribution from both losses, as follows:

$$\mathcal{L}_S = -\frac{1}{D} \sum_{d=1}^{D} \log \left( \sigma \left( \frac{\mathbf{Q}'^{(:,d)}_k \cdot \mathbf{A}^{(:,d)}}{\|\mathbf{Q}'^{(:,d)}_k\| \|\mathbf{A}^{(:,d)}\|} \right) \right) \tag{2}$$

$$\mathcal{L}_{LS} = \frac{1}{2} \left( \mathcal{L}_L + \mathcal{L}_S \right) \tag{3}$$

This formulation ensures that DM-Codec effectively integrates both acoustic and semantic knowledge from SM, along with the contextual information provided by LM, resulting in a more robust and comprehensive set of features for speech discretization.

**[CLS] Token Guided Distillation:** We introduce a *[CLS]-token-based distillation strategy*, leveraging the [CLS] token's sequence-level holistic representation to capture global contextual information

from LM. This approach eliminates the need for fine-grained temporal alignment while preserving essential linguistic features. For this method, we use the layer-wise hidden representations of the [CLS] token alone. These representations, averaged across all layers, are denoted as $\mathbf{S}_{[CLS]} = \frac{1}{L} \sum_{l=1}^{L} \mathbf{h}_{[CLS]}^{l}$, where $\mathbf{S}_{[CLS]} \in \mathbb{R}^D$. To match the sequence length $T'$ of the quantized features $\mathbf{Q}_k'$, the [CLS] token representation is repeated $T'$ times, yielding $\mathbf{S}' = \{\mathbf{S}_{[CLS]}, \mathbf{S}_{[CLS]}, \ldots, \mathbf{S}_{[CLS]}\}$.

The distillation loss follows the same formulation as the LM-guided distillation loss (Eqn. 1), replacing $\mathbf{S}$ with $\mathbf{S}'$. This enables the distillation process to leverage the global contextual information encoded in the [CLS] token while ensuring alignment with the sequence length $T'$.

## 2.2 DM-CODEC: MODEL DETAILS

Our framework builds upon the Residual Vector Quantizer with Generative Adversarial Networks (RVQ-GAN) architecture, incorporating state-of-the-art components and novel distillation techniques. The core of our model consists of an Encoder $\mathbf{E}$ and Decoder $\mathbf{D}$ with an RVQ architecture, inspired by Encodec (Défossez et al., 2022) and SpeechTokenizer (Zhang et al., 2024a). Moreover, we employ a multi-discriminator framework, comprising: Multi-Scale Discriminator (MSD), Multi-Period Discriminator (MPD), and Multi-Scale Short-Time Fourier Transform (MS-STFT) Discriminator, adopted from HiFi-Codec (Yang et al., 2023) and HiFi-GAN (Kong et al., 2020). Detailed architectural specifications for these components are provided in the Appendix D. This foundation provides a robust basis for speech quantization. To further enhance the quantizer with distilled multimodal representations, we use wav2vec 2.0 (wav2vec2-base-960h) as $M_{ASR}$ (Baevski et al., 2020), BERT (bert-base-uncased) as $M_{LM}$ (Devlin et al., 2019), and HuBERT (hubert-base-ls960) as $M_{SM}$ (Hsu et al., 2021). We extract the quantized output from the first layer of the RVQ (RVQ-1) for LM-guided distillation and the average of the quantized features across all eight layers (RVQ-1:8) for SM-guided distillation. Our experiments reveal LM distillation at the RVQ-1 layer best preserves contextual nuances, while SM distillation at the RVQ1:8 layers ensures superior semantic retention. A detailed ablation study of RVQ layers is presented in Appendix C.2.

## 2.3 DM-CODEC: TRAINING OBJECTIVE

Our training strategy employs a GAN-guided framework, following methodologies established in recent work (Zhang et al., 2024a; Yang et al., 2023). In addition to the distillation loss described in Section 2.1, we utilize reconstruction losses, adversarial and feature matching losses, and a commitment loss to guide the learning process. For the original speech $\mathbf{x}$ and the reconstructed speech $\hat{\mathbf{x}}$, we calculate the losses as described below.

**Reconstruction Loss.** To ensure that the model preserves the key attributes of speech, we employ both time-domain and frequency-domain reconstruction losses. The time-domain loss $\mathcal{L}_t$ is computed as the L1 distance between $\mathbf{x}$ and $\hat{\mathbf{x}}$. For the frequency-domain loss $\mathcal{L}_f$, we combine L1 and L2 losses over 64-bin Mel-spectrograms $\text{Mel}_i$, with varying window sizes of $2^i$, hop lengths of $2^i/4$, and scales $e = \{5, \ldots, 11\}$.

$$\mathcal{L}_t = \|\mathbf{x} - \hat{\mathbf{x}}\|_1 \tag{4}$$

$$\mathcal{L}_f = \sum_{i \in e} (\|\text{Mel}_i(\mathbf{x}) - \text{Mel}_i(\hat{\mathbf{x}})\|_1 + \|\text{Mel}_i(\mathbf{x}) - \text{Mel}_i(\hat{\mathbf{x}})\|_2) \tag{5}$$

**Adversarial Loss.** The adversarial loss promotes the generator to produce realistic and indistinguishable speech. We apply a hinge loss formulation to compute the adversarial loss for both the generator $\mathcal{L}_g$ and the discriminator $\mathcal{L}_d$. These losses are computed across all three discriminators: the multi-scale discriminator (MSD), multi-period discriminator (MPD), and the multi-scale STFT guided (MS-STFT) discriminator (details are in the Appendix D).

$$\mathcal{L}_g = \frac{1}{N} \sum_{n=1}^{N} \max(1 - R_n(\hat{\mathbf{x}}), 0) \tag{6}$$

$$\mathcal{L}_d = \frac{1}{N} \sum_{n=1}^{N} (\max(1 - R_n(\mathbf{x}), 0) + \max(1 + R_n(\hat{\mathbf{x}}), 0)) \tag{7}$$

where $N$ is the number of discriminators and $R_n$ represents the $n^{th}$ discriminator.

**Feature Matching Loss.** To prevent the generator from overfitting to the discriminator's decisions, we apply a feature matching loss $\mathcal{L}_{fm}$. This loss compares features from each discriminator $R_n$'s internal layers $M$ across all dimensions, promoting stability and better generalization.

$$\mathcal{L}_{fm} = \frac{1}{NM} \sum_{n=1}^{N} \sum_{m=1}^{M} \frac{\|R_n^m(\mathbf{x}) - R_n^m(\hat{\mathbf{x}})\|_1}{\text{mean}(\|R_n^m(\mathbf{x})\|_1)} \tag{8}$$

**RVQ Commitment Loss.** To guide the encoder to produce outputs that closely match their corresponding quantized values in the residual vector quantization (RVQ) process, we introduce a commitment loss $\mathcal{L}_w$. For $N_q$ quantization vectors, where $\mathbf{q}_i$ represents the current residual and $\mathbf{q}_{c_i}$ is the closest entry in the corresponding codebook for the $i^{th}$ entry, the $\mathcal{L}_w$ is computed as:

$$\mathcal{L}_w = \sum_{i=1}^{N_q} \|\mathbf{q}_i - \mathbf{q}_{c_i}\|_2^2 \tag{9}$$

**Overall Generator Loss.** The total generator loss $\mathcal{L}_G$ is a weighted sum of the individual loss components, including the distillation loss $\mathcal{L}_{L/LS}$ (which is either $\mathcal{L}_L$ or $\mathcal{L}_{LS}$ depending on the chosen distillation method). We use the corresponding weighting factors $\lambda_{L/LS}, \lambda_t, \lambda_f, \lambda_g, \lambda_{fm}$, and $\lambda_w$ to control the influence of each loss component on the overall training objective as:

$$\mathcal{L}_G = \lambda_{L/LS}\mathcal{L}_{L/LS} + \lambda_t\mathcal{L}_t + \lambda_f\mathcal{L}_f + \lambda_g\mathcal{L}_g + \lambda_{fm}\mathcal{L}_{fm} + \lambda_w\mathcal{L}_w \tag{10}$$

This comprehensive training objective ensures DM-Codec learns acoustic speech representations while incorporating semantic and contextual representation through novel distillation approaches.

## 2.4 DM-CODEC-TTS: MODEL DETAILS AND TRAINING OBJECTIVE

Following SpeechTokenizer (Zhang et al., 2024a) and VALL-E (Wang et al., 2023), we propose DM-Codec-TTS, a novel multimodal representation distilled neural codec-based Text-To-Speech (TTS) model. Extending upon general neural codec language models, DM-Codec-TTS, leverages the strength of contextual and semantic representation distilled on our neural codec, DM-Codec.

**Problem Formulation.** For zero-shot TTS, the task is to synthesize speech for a given speaker. We frame it as a conditional codec language modeling problem, where the objective of DM-Codec-TTS is to predict the quantized acoustic features $\mathbf{Q} = \{\mathbf{Q}_k\}_{k=1}^{K}$, conditioned on a phoneme sequence $\mathbf{u}$ and an acoustic prompt $\tilde{\mathbf{P}} \in \mathbb{R}^{T' \times K}$ extracted from the input enrolled recording.

**Training Objective.** The model integrates both autoregressive (AR) and non-autoregressive (NAR) components to hierarchically encode information in speech. The AR component focuses on modeling content and speaker identity by predicting tokens $\mathbf{Q}_1^t$ from the first Residual Vector Quantization (RVQ) layer using a transformer decoder-only architecture $\phi_{\text{AR}}$. The AR training objective is:

$$\mathcal{J}_{\text{AR}} = -\sum_{t=0}^{T} \log p\left(\mathbf{Q}_1^t \mid \mathbf{Q}_1^{<t}, \mathbf{u}; \phi_{\text{AR}}\right)$$

In contrast, the NAR component focuses on acoustic details by predicting tokens $\mathbf{Q}_k$ $(k = 2, \ldots, 8)$ from subsequent RVQ layers. The NAR training objective is:

$$\mathcal{J}_{\text{NAR}} = -\sum_{k=2}^{8} \log p\left(\mathbf{Q}_k \mid \mathbf{Q}_{<k}, \tilde{\mathbf{P}}, \mathbf{u}; \phi_{\text{NAR}}\right)$$

During inference, the AR model predicts tokens $\mathbf{Q}_1$ based on $\mathbf{u}$, while the NAR model iteratively generates $\mathbf{Q}_{2:8}$ using the AR output and the acoustic prompt $\tilde{\mathbf{P}}$. The combined tokens $\mathbf{Q} = \{\mathbf{Q}_1, \mathbf{Q}_{2:8}\}$ are decoded into a speech waveform via the neural codec.

**Model Details.** The AR and NAR models share an identical transformer structure, comprising 12 layers, 16 attention heads, a 1024-dimensional embedding space, 4096-dimensional feed-forward layers, and a dropout rate of 0.1.

## 3 EXPERIMENTAL SETUP

**Dataset.** We trained DM-Codec using the LibriSpeech training set of 100 hours of clean speech (Panayotov et al., 2015). This dataset was selected primarily because of its successful use for training and evaluation in various speech tokenizer and modeling tasks (Zhang et al., 2024a; Ju et al., 2024; Hsu et al., 2021). Before training, we made the data uniform by randomly cropping each sample to three seconds and ensuring a consistent sample rate of 16 Hz. For DM-Codec-TTS, we utilized LibriHeavy (Kang et al., 2024), a 50k-hour dataset of read English speech derived from LibriVox. For training AR and NAR models of DM-Codec-TTS, we selected speech samples with durations between 0.5 and 15 seconds; all data are sampled at 16 kHz. We also trained a DM-Codec-TTS version, termed DM-Codec-TTS-small, on LibriTTS (Zen et al., 2019), a dataset comprising 585 hours of speech data sampled at 24 kHz from 2,456 speakers, along with the corresponding texts.

**Training.** We trained DM-Codec utilizing 2 to 4 A100 GPUs until the model converged within 100 epochs. The batch size ranged from 6 to 20, depending on GPU resource availability. We applied a learning rate of $1 \times 10^{-4}$ using the Adam optimizer with a 0.98 learning rate decay. The embedding size was set to 1024 for RVQ and 768 for the LM and SM. For all experiments, we used a random seed of 42 to ensure reproducibility. For the overall generator loss, we select the weight coefficients proportionally as follows: $\lambda_{L/LS} = X$, $\lambda_f = 0.375X$, $\lambda_t = 4.15X$, $\lambda_w = 0.085X$, and set $\lambda_g = 1$ and $\lambda_{fm} = 1$. For DM-Codec-TTS, we trained the autoregressive (AR) and non-autoregressive (NAR) models separately, each for 4 epochs, while DM-Codec-TTS-small was trained with 100 epochs for the AR model and 196 epochs for the NAR model. The batch size was determined dynamically based on the maximum number of audio seconds, set to 280 for AR and 200 for NAR. We employed a base learning rate of 0.05 using the ScaledAdam optimizer with warmup steps of 200. We also share our training code with the entire configuration file and a docker file to reproduce the training environment in the Appendix A.

**Baselines.** We compared DM-Codec with the baseline speech tokenizers: EnCodec (Défossez et al., 2022), SpeechTokenizer (Zhang et al., 2024a), and FACodec (NaturalSpeech3) (Ju et al., 2024). We reproduced SpeechTokenizer using the official training code and used official model checkpoints of EnCodec and FACodec as the baselines. Additionally, we compared DM-Codec-TTS with neural codec language models, USLM (from SpeechTokenizer (Zhang et al., 2024a)) and VALL-E (Wang et al., 2023). As VALL-E and USLM's official training codes and models are not open-source, we relied on results reported in their respective papers. For DM-Codec-TTS-small, we used USLM (libri) as the baseline, an official model checkpoint trained on LibriTTS and shared via GitHub.

**Evaluation Dataset.** To evaluate DM-Codec, we randomly selected 300 audio samples from the LibriSpeech test subset, following a similar practice of sampling test data used in our baselines (Zhang et al., 2024a; Zeghidour et al., 2021) and to align the experimental setup with that of Speech-Tokenizer. In our experiments, we sampled the test subset of LibriSpeech using a random seed of 42. We also evaluated the baseline models with the same sampled test dataset for a fair comparison. We evaluated DM-Codec-TTS on two distinct datasets: LibriSpeech test-clean subset, featuring 40 speakers, and VCTK, featuring 110 speakers. To align with VALL-E's (Wang et al., 2023) experimental setup, we constructed a 2.2-hour subset from LibriSpeech test-clean, selecting samples with durations between 4 and 10 seconds. During synthesis, a separate utterance from the same speaker was randomly chosen, and a 3-second segment was cropped as a reference speech for enrollment. For consistency and reliability, each experiment was repeated three times, and the average score was reported. Following the setup of SpeechTokenzier (Zhang et al., 2024a) for VCTK, we selected a 3-second utterance from each speaker to serve as the prompt. A separate utterance from the same speaker, containing different textual content, was used as input text for synthesis.

**Evaluation Metrics.** We employed different metrics to compare and evaluate the reconstructed speech from DM-Codec and synthesized speech from DM-Codec-TTS. First, we used the Word Error Rate (WER) and Word Information Lost (WIL) metrics to evaluate context preservation by calculating the amount of word-level transcription errors and key information missing in transcription, respectively. For these metrics, we used the Whisper (whisper-medium) (Radford et al., 2023)

model to extract the transcription from the reconstructed speech. To provide a fairer comparison and indicate the level of transcription error by the Whisper model, we also included the Groundtruth WER and WIL scores for the Whisper's transcribed text from the original speech versus the true text. Next, we assessed the acoustic and semantic information preservation in reconstructed speech using the ViSQOL (Virtual Speech Quality Objective Listener) (Hines et al., 2012) and Short-Time Objective Intelligibility (STOI) metrics, respectively. The ViSQOL metric measures the similarity between a reference and a test speech sample using a spectro-temporal measure and produces a MOS-LQO (Mean Opinion Score - Listening Quality Objective) score ranging from 1 (worst) to 5 (best). For this metric, we used the wideband model suited for speech evaluation. Lastly, the STOI metric evaluates the perceived intelligibility of speech by analyzing short-time correlations between original and reconstructed speech, with scores ranging from 0 to 1.

We conducted human evaluations to measure the Mean Opinion Score (MOS) and Similarity Mean Opinion Score (SMOS) using 50 English-proficient participants. Evaluations used randomized, anonymized samples rated on a 1-to-5 scale, with higher scores indicating better performance. MOS assesses the naturalness, intelligibility, and clarity of reconstructed and synthesized speech, while SMOS evaluates similarity to the prompt speaker's voice. Additionally, speaker similarity (Similarity) between the synthesized speech and the prompt speech was quantified using cosine similarity between normalized speaker embeddings extracted with WavLM-TDNN (Chen et al., 2022). Details on human evaluations are in Appendix G.

## 4 EXPERIMENTAL RESULTS AND DISCUSSION

To evaluate the performance of DM-Codec, we conducted a comprehensive set of experiments assessing speech reconstruction quality and contextual retention. Our analysis compared variants of DM-Codec —DM-Codec (LM), employing LM-guided distillation, DM-Codec (LM+SM), incorporating both LM and SM-guided distillation, and DM-Codec (CLS), which integrates [CLS]-token guided distillation—against state-of-the-art (SOTA) speech tokenization models. Moreover, we compare DM-Codec-TTS and DM-Codec-TTS-small with SOTA neural codec language models.

### 4.1 SPEECH RECONSTRUCTION EVALUATION

Table 1: Evaluation of speech reconstruction quality of DM-Codec and comparison with baselines. DM-Codec (LM+SM) achieves the best performance in WER, WIL, ViSQOL, and MOS, highlighting its enhanced content preservation and speech quality. $\heartsuit$ means the results were reproduced using the official training code. $\diamond$ means the results were obtained using official model checkpoints. (LM) indicates LM-guided distillation method. (LM+SM) indicates combined LM and SM-guided distillation method. CLS indicates [CLS]-token based distillation method. Baseline indicates DM-Codec without any distillation method. **Bold** highlights the best and underline the second-best result.

| Model | WER ↓ | WIL ↓ | ViSQOL ↑ | STOI ↑ | Similarity ↑ | MOS ↑ |
|---|---|---|---|---|---|---|
| Groundtruth | 3.78 | 6.03 | - | - | - | - |
| EnCodec$^{\diamond}$ | 4.53 | 7.17 | 3.08 | 0.920 | 0.980 | 3.09 |
| SpeechTokenizer$^{\heartsuit}$ | 4.49 | 7.10 | 3.09 | 0.923 | 0.993 | 3.67 |
| FACodec$^{\diamond}$ | 4.68 | 7.33 | 3.13 | **0.949** | **0.996** | 3.70 |
| DM-Codec (Baseline) | 4.97 | 8.02 | 2.95 | 0.935 | 0.991 | 3.13 |
| DM-Codec (LM) | 4.36 | 7.06 | 3.18 | 0.935 | 0.994 | 3.69 |
| DM-Codec (LM+SM) | **4.05** | **6.61** | **3.26** | 0.937 | 0.994 | **3.72** |
| DM-Codec (CLS) | 4.47 | 7.08 | 3.12 | 0.926 | 0.993 | 3.65 |

We compared the quality of DM-Codec's discrete speech representations by reconstructing speech from quantized vector features and comparing it with state-of-the-art speech tokenization models: EnCodec, SpeechTokenizer, and FACodec. For this evaluation, we select DM-Codec variants where the first Residual Vector Quantizer layer (RVQ-1) was used for LM distillation and all RVQ layers (RVQ-1:8) were employed for SM distillation.

**Results:** The performance results, summarized in Table 1, demonstrate that all variants of DM-Codec either surpass or closely compete with the baselines. Specifically, DM-Codec (LM) outperforms baselines on content preservation (WER 4.36, WIL 7.06, and ViSQOL 3.18) and EnCodec and SpeechTokenier on speech quality (ViSQOL 0.935, STOI 0.994, MOS 3.69). DM-Codec (LM+SM) with combined LM and SM-guided distillation outscores DM-Codec (LM) and all previous scores

with 4.05 WER, 6.61 WIL, 3.26 ViSQOL, and 3.72 MOS, while acquiring second best 0.937 STOI and 0.994 Similarity scores. In addition, DM-Codec (CLS) with [CLS]-token based distillation also outperforms the baselines in terms of WER (4.47) and WIL (7.08), and maintains a compatible ViSQOL (3.12), STOI (0.926), Similarity (0.993), and MOS (3.65) scores.

**Discussion:** The superior performance of DM-Codec (LM) is attributed to its innovative LM-guided distillation, which incorporates contextual representations into the model. This approach enhances the global alignment of speech features with contextual cues, leading to significant reductions in WER and WIL, as well as improvements in speech characteristics, as evidenced by the strong ViSQOL, STOI, and MOS scores. The combined LM- and SM-guided distillation in DM-Codec (LM+SM) further amplifies these benefits by integrating semantic representations into the model. This dual representation—contextual alignment from LM and semantic understanding from SM—enables a more coherent and natural reconstruction of speech, yielding superior results across all evaluation metrics. Additionally, the [CLS]-token-guided distillation offers a holistic representation of the entire contextual input, facilitating enhanced alignment of contextual cues. This method's ability to retain meaningful content is reflected in its competitive WER and WIL scores and balanced performance across other metrics. The impact of these distillation techniques becomes evident when compared to the DM-Codec (Baseline) model, which does not use any distillation and demonstrates significantly lower performance across all metrics. This highlights the critical role of distillation in enhancing both the contextual and semantic dimensions of speech representation.

Table 2: Significance Analysis on LibriSpeech Test Set. Significance analysis is conducted at $\alpha = 0.05$ between LM and SM-guided **DM-Codec (D), EnCodec (E), SpeechTokenizer (S), and FACodec (F)**. Comparisons are performed row vs. column (e.g., D vs. E, E vs. S). Results reveal DM-Codec consistently achieves significantly better scores in key metrics across all individual samples. A ✓ indicates significance, a ★ denotes dominance, and a ✗ means no significance. Avg and Std mean the average and standard deviation of each score.

| | WER ↓ | | | | | | WIL ↓ | | | | | | ViSQOL ↑ | | | | | | STOI ↑ | | | | | |
|---|---|---|---|---|---|---|---|---|---|---|---|---|---|---|---|---|---|---|---|---|---|---|---|---|
| | Avg | Std | D | E | S | F | Avg | Std | D | E | S | F | Avg | Std | D | E | S | F | Avg | Std | D | E | S | F |
| D | 4.774 | 0.100 | - | ✓ | ✓ | ✓ | 7.510 | 0.139 | - | ✓ | ✓ | ✓ | 3.197 | 0.184 | - | ★ | ✓ | ✓ | 0.937 | 0.021 | - | ✓ | ✓ | ✗ |
| E | 4.828 | 0.100 | ✗ | - | ✓ | ✓ | 7.593 | 0.137 | ✗ | - | ✓ | ✓ | 3.064 | 0.201 | ✗ | - | ✗ | ✗ | 0.917 | 0.021 | ✗ | - | ✗ | ✗ |
| S | 4.942 | 0.101 | ✗ | ✗ | - | ✗ | 7.725 | 0.138 | ✗ | ✗ | - | ✗ | 3.080 | 0.190 | ✗ | ✓ | - | ✗ | 0.920 | 0.025 | ✗ | ✓ | - | ✗ |
| F | 4.914 | 0.103 | ✗ | ✗ | ✓ | - | 7.643 | 0.141 | ✗ | ✗ | ✓ | - | 3.113 | 0.250 | ✗ | ✓ | ✓ | - | 0.946 | 0.023 | ✓ | ✓ | ✓ | - |

## 4.2 SIGNIFICANCE ANALYSIS OF SPEECH TOKENIZER PERFORMANCE

We conducted a significance analysis at $\alpha = 0.05$ on the LibriSpeech test-clean subset containing 2,620 samples. We follow the approach of Dror et al. (2019), to measure the stochastic dominance of DM-Codec over the baselines: EnCodec, SpeechTokenizer, and FACodec. Specifically, we computed inverse cumulative distribution functions (CDFs) for all reconstructed speech samples' individual WER, WIL, ViSQOL, and STOI scores. Significance was evaluated using the $\epsilon$ value and categorized as: significantly better when $0.0 < \epsilon \leq 0.5$, significantly dominant when $\epsilon = 0.0$, and not significantly better when $\epsilon > 0.5$. For this analysis, we selected DM-Codec (LM+SM), trained with combined LM and SM-guided distillation. **To the best of our knowledge, we are the first to conduct significance analysis to measure the effectiveness of different speech tokenizers.**

**Results and Discussion:** The results in Table 2 show that DM-Codec significantly outperforms the baselines in WER, WIL, ViSQOL, and STOI scores. The improved average values (4.774 WER, 7.510 WIL, 3.197 ViSQOL, 0.937 STOI) and consistent standard deviations (0.100 WER, 0.139 WIL, 0.184 ViSQOL, 0.021 STOI) further demonstrate the statistical significance. Notably, DM-Codec's performance in WER and WIL underscores the importance of contextual representation distillation for enhanced speech reconstruction. Additionally, its strong performance in ViSQOL and STOI, especially over EnCodec, highlights the benefits of combining LM and SM distillation for retaining semantic-acoustic fidelity. While DM-Codec does not achieve significance over FA-Codec in terms of STOI, it significantly outperforms the baselines across all other metrics. Among the baseline, FACodec achieves significance over SpeechTokenizer, whereas EnCodec outperforms SpeechTokenizer in WER and WIL, SpeechTokenizer excels in ViSQOL and STOI over EnCodec.

Table 3: Evaluation of DM-Codec-TTS on the LibriSpeech and VCTK datasets. Results show that DM-Codec-TTS outperforms other models in terms of WER, WIL, Similarity, MOS, and SMOS. $^{\diamond}$ denotes results obtained using officially shared model checkpoints, and $^{\dagger}$ indicates results directly obtained from the paper. ♣ indicates model trained with LibriTTS dataset.

| | LibriSpeech Evaluation | | | | | VCTK Evaluation | | | | |
|---|---|---|---|---|---|---|---|---|---|---|
| **Model** | **WER ↓** | **WIL ↓** | **Similarity ↑** | **MOS ↑** | **SMOS ↑** | **WER ↓** | **WIL ↓** | **Similarity ↑** | **MOS ↑** | **SMOS ↑** |
| DM-Codec-TTS | **5.08** | **7.32** | **0.82** | **3.70** | 3.89 | **3.58** | **5.65** | 0.82 | **3.78** | **3.85** |
| Vall-E $^{\dagger}$ | 5.90 | - | 0.58 | - | **4.38** | - | - | 0.38 | - | 3.81 |
| USLM $^{\dagger}$ | - | - | - | - | - | 6.50 | - | **0.84** | 3.63 | 3.45 |
| DM-Codec-TTS♣ | 10.26 | 13.79 | 0.82 | 3.24 | 3.20 | 5.02 | 8.21 | 0.79 | 3.39 | 3.28 |
| USLM_libri $^{\diamond}$ ♣ | 16.72 | 25.65 | 0.80 | 3.11 | 2.83 | 14.79 | 23.24 | 0.78 | 2.94 | 2.63 |

### 4.3 SPEECH SYNTHESIS TEXT-TO-SPEECH EVALUATION

We compared the Zero-shot TTS ability of DM-Codec-TTS with baseline USLM and VALL-E. We use DM-Codec (LM+SM) to encode acoustic prompt features as input and decode quantized acoustic features predicted by DM-Codec-TTS.

**Results and Discussion.** The results in Table 3 show that DM-Codec-TTS outperforms USLM and VALL-E baselines. In both benchmark evaluations, DM-Codec achieves the lowest WER (5.08, 3.58), WIL (7.32, 5.65), and highest MOS (3.70, 3.78), while achieving closely aligned Similarity to USLM in VCTK, and superior SMOS 3.85 compared to VALL-E and USLM in VCTK. Moreover, DM-Codec-small significantly outperforms USLM in all metrics in both benchmarks.

The improved performance strongly indicates that hierarchical modeling in DM-Codec-TTS effectively utilizes the distilled contextual and semantic knowledge in DM-Codec. This highlights the benefits of hierarchical modeling to bridge the gap between linguistic content and acoustic fidelity. The improved performance strongly indicates that hierarchical modeling in DM-Codec-TTS effectively utilizes the contextual and semantic knowledge distilled in DM-Codec. Unlike VALL-E, which relies on EnCodec to model audio tokens, DM-Codec-TTS leverages distilled multimodal representation cues to integrate fine-grained acoustic details with high-level semantic understanding. This demonstrates the strength of contextual and semantic-aware hierarchical modeling in bridging linguistic content with acoustic fidelity, leading to more natural and intelligible speech synthesis.

## 5 RELATED WORK

The adoption of textual LMs for speech-related tasks is a promising direction. Generally, an audio encoder converts audio signals into discrete representations, which are passed to pre-trained textual LLMs. This approach has been explored by (Hassid et al., 2024), (Wang et al., 2024), (Zhang et al., 2023), (Fathullah et al., 2023), (Shu et al., 2023), and (Rubenstein et al., 2023). Another method involves the corresponding text to feed directly into an LM (Zhang et al., 2024b). Most of these approaches aim to extract representations through LMs while focusing on speech reconstruction training objectives. Recently, LAST (Turetzky & Adi, 2024), explored a language model to tokenize speech toward improved sequential modeling, using the LLM to perform the next token prediction of quantized vectors. However, these approaches significantly differ from our method and do not focus on combining multimodal representations. More details are reported in the Technical Appendix E.

## 6 CONCLUSION

In this work, we introduced DM-Codec, a speech tokenizer with novel distillation methods leveraging multimodal (acoustic, semantic, and contextual) representations from language and speech self-supervised models. Experimental results and ablation studies show that distilling multimodal representations enables DM-Codec to introduce salient speech information in discrete speech tokens. Our significance analysis further revealed that DM-Codec with comprehensive multimodal representations consistently outperforms existing speech tokenizers. This approach highlights the potential of multimodal representations to enhance speech tokenization in various domains, including multilingual and code-switched speech processing.

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

# Technical Appendix

## A    RESOURCES

We provide the code for training DM-Codec, trained model checkpoints, and Dockerfile for a reproducible code environment. The links are shared anonymously for the double-blind review process. We will publicly share all resources after the completion of the review timeline.

- **Model training codebase**: Codebase
- **Trained model checkpoints for inference**: Model-checkpoints
- **Dockerfile for reproducible environment**: Docker

## B    REPRESENTATION ALIGNMENT IN LM DISTILLATION

A core principle of our approach is that strict temporal alignment between text and acoustic representations is not necessary for effective contextual knowledge distillation. The discrete representations produced by the Residual Vector Quantizer (RVQ) inherently encode holistic information about the speech segment rather than temporally localized features. This characteristic enables the distillation of contextual representations into discrete tokens without reliance on strict temporal correspondence.

As illustrated in Figure 2, the RVQ outputs $\{Q_1, \ldots, Q_{T'}\}$ capture comprehensive speech information across the entire utterance. Temporal alignment between these discrete representations and contextual representations (e.g., from BERT) or semantic representations (e.g., from HuBERT) is therefore not essential. The effectiveness of dimension-level semantic representation distillation without temporal alignment has been previously demonstrated in SpeechTokenizer (Zhang et al., 2024a), providing a robust theoretical foundation for this approach.

Vector quantizer output representations are not inherently aligned with time steps, but instead encode holistic speech information. This capability has been established in prior work (Zhang et al., 2024a; Huijben et al., 2024; Islam et al., 2024; 2023), which demonstrates the potential of vector quantization to discretize input data into intermediate representations that capture essential features across the feature dimension. Consequently, imposing a temporal alignment between vector quantizer outputs and hidden layer representations from language or semantic models would neither align with the methodological objectives nor enhance the efficacy of the proposed distillation approach.

To achieve effective contextual and semantic knowledge transfer, we employ a continuous distillation loss that **maximizes cosine similarity at the feature dimension level** between the selected RVQ layer outputs and the teacher representations across all time steps. Unlike conventional methods that rely on time-step-wise loss calculations (Zhang et al., 2024a; Chang et al., 2022), this dimension-level cosine similarity loss ensures that DM-Codec captures contextual and semantic knowledge through LM-Guided Distillation and Combined LM and SM-Guided Distillation mechanisms, without requiring strict temporal alignment.

In addition, we propose a **[CLS]-token-based distillation strategy** to address alignment concerns. The [CLS] token encodes sequence-level holistic representations, capturing global contextual information from language models. By leveraging this token, our method eliminates the need for fine-grained temporal alignment while preserving essential linguistic features. This complements the dimension-level distillation strategy by focusing on global sequence features, enabling the adaptability of our approach to scenarios where fine-grained alignment is infeasible or unnecessary.

To validate the effectiveness of the **[CLS]-token-based distillation strategy**, additional experiments were conducted. As shown in Table 1, all DM-Codec variants—including DM-Codec (CLS), DM-Codec (LM-guided), and DM-Codec (LM+SM-guided)—consistently outperform baseline models (EnCodec, SpeechTokenizer, and FACodec) in content preservation metrics (WER and WIL) while maintaining competitive performance in speech quality metrics (ViSQOL and STOI). These results corroborate the robustness of the proposed approach.

By prioritizing the holistic integration of multimodal knowledge into discrete speech representations, rather than strict temporal alignment, DM-Codec achieves significant advancements in both content preservation and speech quality.

## C    ABLATION STUDIES

We conducted a comprehensive analysis of DM-Codec's performance and the impact of each methodological choice in LM-guided and combined LM and SM-guided distillation. Unless otherwise stated, we use distillation for both LM and SM to the first Residual Vector Quantizer layer (RVQ-1) for comparison consistency and simplicity. The following ablation studies were conducted concurrently with a similar configuration and model design except for the explicitly noted changes.

### C.1    ABLATION STUDY: IMPACT OF COMBINED SEMANTIC DISTILLATION

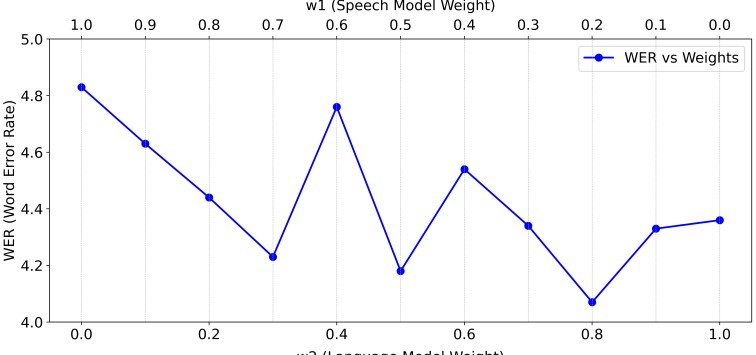

Figure 3: Effects of weights on combined distillation from Speech Model (SM) and Language Model (LM). Higher LM weight generally results in improved WER, suggesting its stronger contribution to content preservation. Here, $\lambda_{SM}$ is the weight added to SM and $\lambda_{LM}$ is the weight added to LM, where ($\lambda_{SM} + \lambda_{LM} = 1$).

We conducted experiments with different weighted combinations of LM and SM distillation loss to evaluate their impact on reducing WER. The combined distillation loss from Equation 3 was updated using SM and LM weights ($\lambda_{SM}$ and $\lambda_{LM}$), ranging from $0.0$ to $1.0$, with the constraint $\lambda_{SM} + \lambda_{LM} = 1$.

$$\mathcal{L}_{LS} = \frac{1}{2} \left( \lambda_{LM} \cdot \mathcal{L}_L + \lambda_{SM} \cdot \mathcal{L}_S \right) \tag{11}$$

**Results and Discussion:** The experimental results are presented in Figure 3, showing the speech reconstruction results with WER scores for different weighted combinations. From the values, we notice a trend showing that incorporating LM representations generally improves WER, especially when LM distillation is dominant. The lowest WER score of 4.07 occurs with a weight of $\lambda_{LM} = 0.8$ for LM, while $\lambda_{SM} = 0.2$ for SM, highlighting the strong influence of LM distillation on capturing contextual information. A balanced weighting of $\lambda_{SM} = 0.5$ and $\lambda_{LM} = 0.5$ produces a WER of 4.18, confirming that distillation from both LM and SM is beneficial. However, as the weighting shifts more in favor of SM ($\lambda_{SM} > 0.7$), WER deteriorates, reaching 4.83 when relying entirely on SM. This underscores that over-reliance on SM distillation compromises contextual accuracy in favor of raw speech features. Notably, the interaction between LM and SM weights plays a crucial role, as the combined distillation influences the overall WER beyond individual distillation contributions. For instance, the higher WER observed at $\lambda_{LM} = 0.9$ compared to $\lambda_{LM} = 0.3$ or $\lambda_{LM} = 0.5$ highlights the importance of tuning both weights synergistically, rather than favoring one in isolation.

### C.2    ABLATION STUDY: IMPACT OF DISTILLATION ON DIFFERENT RVQ LAYERS

We evaluated the effect of applying distillation at various Residual Vector Quantizer (RVQ) layers, including the first layer (RVQ-1), the average of eight layers (RVQ-1:8), and the last layer (RVQ-8). Table 4 shows the full results.

**Results and Discussion:** In LM-guided distillation, RVQ-1:8 achieves the best WER and WIL scores (4.23 and 6.94), though with lower ViSQOL and STOI scores (3.12 and 0.929) compared to

Table 4: Analysis of different RVQ layers effect on speech reconstruction. LM-guided distillation on RVQ-1 layer ensures greater content preservation, while SM-guided distillation on RVQ-1:8 layer is more effective at preserving semantic representation. LM-layer and SM-layer indicate the RVQ layer used for respective distillation. (LM) indicates LM-guided Distillation. (LM+SM) indicates combined LM and SM-guided Distillation. **Bold** highlights the best result and underline the second-best result.

| Tokenizer | LM-Layer | SM-Layer | WER ↓ | WIL ↓ | ViSQOL ↑ | STOI ↑ |
|---|---|---|---|---|---|---|
| DM-Codec (LM) | RVQ-1 | - | 4.36 | 7.06 | 3.18 | 0.935 |
| DM-Codec (LM) | RVQ-1:8 | - | 4.23 | 6.94 | 3.12 | 0.929 |
| DM-Codec (LM) | RVQ-8 | - | 4.44 | 7.22 | 3.28 | 0.935 |
| DM-Codec (LM+SM) | RVQ-1 | RVQ-1 | 4.18 | 6.84 | 3.13 | 0.933 |
| DM-Codec (LM+SM) | RVQ-1:8 | RVQ-1 | 4.59 | 7.34 | 3.21 | 0.937 |
| DM-Codec (LM+SM) | RVQ-8 | RVQ-1 | 4.49 | 7.24 | 3.30 | 0.938 |
| DM-Codec (LM+SM) | RVQ-1 | RVQ-1:8 | **4.05** | **6.61** | 3.26 | 0.937 |
| DM-Codec (LM+SM) | RVQ-1 | RVQ-8 | 4.39 | 7.08 | **3.33** | **0.939** |

RVQ-8 (3.28 and 0.935). The RVQ-1 layer provides the best overall balance between content preservation and perceptual quality, with WER, WIL, ViSQOL, and STOI scores of 4.36, 7.06, 3.18, and 0.935. This demonstrates RVQ-1:8 prioritizes contextual integrity, while RVQ-8 favors perceptual quality. Thus, we select RVQ-1 for LM-guided distillation due to its balanced performance.

For LM and SM-based distillation, the RVQ-1 and RVQ-1:8 combination achieves the best WER and WIL scores (4.05 and 6.61), with RVQ-1 and RVQ-1 as the second-best (4.18 and 6.84). In contrast, the RVQ-1 and RVQ-8 combination yields the highest ViSQOL and STOI scores (3.33 and 0.939), followed by RVQ-8 and RVQ-1 (3.30 and 0.938). RVQ-1 captures contextual representation more effectively due to its simpler quantized vector, while RVQ-1:8 incorporates more nuanced semantic and acoustic aspects. Overall, this ablation shows that selecting RVQ layers for LM and SM-based distillation greatly affects the balance between contextual accuracy and semantic-acoustic fidelity, allowing layer combinations to be tailored to task requirements.

## C.3 ABLATION STUDY: IMPACT OF DIFFERENT MODELS ON DISTILLATION

Table 5: Analysis of representation distillation from different models. BERT can be effectively combined with HuBERT or wav2vec 2.0, however, ELECTRA in LM-guided distillation outperforms BERT. (LM) indicates LM-guided Distillation. (LM+SM) indicates combined LM and SM-guided Distillation. **Bold** highlights the best result and underline the second-best result.

| Tokenizer | LM | SM | WER ↓ | WIL ↓ | ViSQOL ↑ | STOI ↑ |
|---|---|---|---|---|---|---|
| DM-Codec (LM) | BERT | - | 4.36 | 7.06 | 3.18 | 0.935 |
| DM-Codec (LM) | ELECTRA | - | **4.12** | **6.63** | 3.10 | 0.936 |
| DM-Codec (LM+SM) | BERT | HuBERT | 4.18 | 6.84 | 3.13 | 0.933 |
| DM-Codec (LM+SM) | BERT | wav2vec 2.0 | 4.13 | 6.77 | **3.15** | **0.942** |
| DM-Codec (LM+SM) | ELECTRA | wav2vec 2.0 | 4.70 | 7.51 | 3.14 | 0.933 |
| DM-Codec (LM+SM) | ELECTRA | HuBERT | 4.67 | 7.58 | 2.94 | 0.932 |

We experimented with different LM and SM distillations to analyze performance variations based on different model selections. In addition to our selected BERT (Devlin et al., 2019) and HuBERT (Hsu et al., 2021), we experiment with ELECTRA (electra-base-discriminator) (Clark et al., 2020) as the LM and wav2vec 2.0 (wav2vec2-base-960h) (Baevski et al., 2020) as the SM. Table 5 shows the full results.

**Results and Discussion:** In LM-guided distillation, the ELECTRA model significantly enhances performance, achieving WER and WIL scores of 4.12 and 6.63, respectively, compared to BERT's scores of 4.36 and 7.06. This indicates the architecture of ELECTRA's effectiveness for the proposed LM-guided distillation, demonstrating its superior contextual representation. These results are consistent with ELECTRA's better performance in general natural language processing tasks. However, we select BERT for its simplicity and established performance.

In LM and SM-guided distillation, the combination of BERT and wav2vec 2.0 achieves the highest overall performance, with scores of WER 4.13, WIL 6.77, ViSQOL 3.15, and STOI 0.942. However, the combination of BERT and HuBERT closely follows with second-best scores of WER 4.18, WIL 6.84, and ViSQOL 0.933. These findings demonstrate that different speech models can be effectively integrated with the BERT model.

## C.4 ABLATION STUDY: IMPACT OF DIFFERENT DISTILLATION LAYER(S)

Table 6: Analysis of different distillation layers representation on speech reconstruction. Average layer provides more comprehensive representations. (LM) indicates LM-guided Distillation. (LM+SM) indicates combined LM and SM-guided Distillation. **Bold** highlights the best result and underline the second-best result.

| Tokenizer | Distillation Layer(s) | WER ↓ | WIL ↓ | ViSQOL ↑ | STOI ↑ |
|---|---|---|---|---|---|
| DM-Codec (LM) | Average | 4.36 | 7.06 | **3.18** | **0.935** |
| DM-Codec (LM) | Last | 4.62 | 7.56 | 2.95 | 0.926 |
| DM-Codec (LM) | $9^{th}$ | 4.75 | 7.80 | 2.88 | 0.925 |
| DM-Codec (LM+SM) | Average | **4.18** | **6.84** | 3.13 | 0.933 |
| DM-Codec (LM+SM) | Last | 4.68 | 7.55 | 3.03 | 0.933 |
| DM-Codec (LM+SM) | $9^{th}$ | 4.52 | 7.43 | 3.00 | 0.933 |

We evaluated speech reconstruction using different distillation layers of the LM and SM, examining which combination of layers yields the most relevant representations of semantic and contextual information. For this ablation, we considered the average of all layer representations, the $9^{th}$ layer representations, and the last layer representations. Table 6 shows the full results.

**Results and Discussion:** In LM-guided distillation, the use of the average layer achieves superior overall performance, with a WER of 4.36, WIL of 7.06, ViSQOL of 3.18, and STOI of 0.935, compared to the variants utilizing the last and 9th layers. Similarly, in LM and SM-guided distillation, the average layer yields superior results compared to the last and $9^{th}$ layer variants.

The results indicate that averaging all layers leads to more comprehensive representations of semantic or contextual information. In the case of LM, the averaging process provides greater contextual representation and synergizes syntactic information from earlier layers and abstract word relations from higher layers. In combined LM and SM-guided distillation, averaging all SM layers provides a more nuanced understanding of the earlier layer's phonetic information and the higher layers' richer semantic information. Conversely, relying solely on the last layer or the $9^{th}$ layer fails to capture the overall context and semantic information, yielding less relevant representation distillation.

## C.5 ABLATION STUDY: IMPACT OF LOW BIT RATE (S)

Table 7: Low Bit Rate Speech Reconstruction Evaluation. DM-Codec (LM+SM) showcases its robustness at reduced bitrates, outperforming baselines at 3 kbps and maintaining competitive content preservation scores (WER, WIL) and superior speech quality (ViSQOL, STOI) at 1.5 kbps. $f_s$ represents the audio sample rate, and $f_r$ the codec frame rate. ♡ means the results were reproduced using the official training code. ◇ means the results were obtained using official model checkpoints. (LM) indicates LM-guided Distillation method. (LM+SM) indicates combined LM and SM-guided Distillation method.

| Model | $f_s$ | $f_r$ | Bitrate | WER ↓ | WIL ↓ | ViSQOL ↑ | STOI ↑ |
|---|---|---|---|---|---|---|---|
| DM-Codec (LM+SM) | 16 kHz | 50 Hz | 3 kbps | **4.29** | **7.04** | **3.070** | **0.928** |
| DM-Codec (LM) | 16 kHz | 50 Hz | 3 kbps | 4.38 | 7.09 | 3.042 | 0.924 |
| SpeechTokenizer♡ | 16 kHz | 50 Hz | 3 kbps | 4.70 | 7.43 | 2.905 | 0.911 |
| EnCodec◇ | 24 kHz | 50 Hz | 3 kbps | 4.80 | 7.80 | 2.550 | 0.872 |
| DM-Codec (LM) | 16 kHz | 50 Hz | 1.5 kbps | 6.14 | 10.13 | 2.644 | 0.880 |
| DM-Codec (LM+SM) | 16 kHz | 50 Hz | 1.5 kbps | 6.19 | 10.16 | **2.662** | **0.894** |
| SpeechTokenizer♡ | 16 kHz | 50 Hz | 1.5 kbps | **5.61** | **9.02** | 2.500 | 0.846 |
| EnCodec◇ | 24 kHz | 50 Hz | 1.5 kbps | 10.53 | 16.63 | 2.443 | 0.809 |

We evaluated speech reconstruction with reduced bitrates of 1.5kbps and 3kbps in DM-Codec and compared it with SpeechTokenizer and EnCodec. Both DM-Codec and SpeechTokenizer are trained on 16kHz sample rates and produce 50Hz codec frame rates, whereas EnCodec is trained on 24kHz sample rates and also produces 50Hz codec frame rates. To reduce the bitrate of DM-Codec and SpeechTokenizer, we limited RVQ levels, specifically keeping the first 3 RVQ layers to achieve 1.5kbps and 6 RVQ layers for 3kbps. For EnCodec, we kept the first two VQ layers for 1.5kbps and 4 VQ layers for 3kbps. Table 7 shows the full results.

**Results and Discussion:** At a 3kbps bitrate, LM-guided DM-Codec (LM) maintains its performance and consistency, surpassing the baseline with scores of 4.38 WER, 7.09 WIL, 3.042 ViSQOL, and 0.924 STOI. Combined LM and SM-guided DM-Codec (LM+SM) further improves these scores to 4.29 WER, 7.04 WIL, 3.070 ViSQOL, and 0.928 STOI, outperforming all baselines.

At 1.5kbps, both DM-Codec (LM) and DM-Codec (LM+SM) slightly lag behind SpeechTokenizer in WER and WIL scores but maintain excellent speaker quality, achieving the best ViSQOL and STOI scores of (2.644, 0.880) and (2.662, 0.894), respectively. We hypothesize that the performance degradation in WER and WIL is due to the loss of contextual representation at lower bitrates, as the reduced bandwidth limits the model's ability to capture and preserve nuanced contextual details incorporated into DM-Codec through distillation.

## D MODEL COMPONENTS

**Encoder Decoder.** The encoder-decoder architecture in DM-Codec is based on SEANet (Tagliasacchi et al., 2020), leveraging the successful design employed in recent speech tokenization models (Zhang et al., 2024a; Défossez et al., 2022; Zeghidour et al., 2021). The architecture is designed to efficiently process and reconstruct speech signals while maintaining high fidelity. The Encoder $\mathbf{E}$ consists of a 1D convolution layer with $C$ channels and a kernel size of 7, followed by $B$ residual convolutional blocks. Each block contains a strided convolutional downsampling layer with kernel size $K$ (where $K = 2S$, and $S$ represents the stride), paired with a residual unit. The residual unit comprises two convolutional layers with a kernel size of 3 and a skip connection, while the number of channels is doubled at each downsampling stage. This is followed by a two-layer BiLSTM and a final 1D convolutional layer with $D$ output channels and a kernel size of 7. The Decoder $\mathbf{D}$ mirrors the encoder's structure but replaces BiLSTM with LSTM, strided convolutions with transposed convolutions, and employs reversed strides for up-sampling. The final audio output is reconstructed from $\mathbf{D}$. For the experiments, we use the following configuration: $C = 32$, $B = 4$, and $S = (2, 4, 5, 8)$.

**Residual Vector Quantizers.** The Residual Vector Quantizer (RVQ) plays a central role in our tokenization process, quantizing the encoder's outputs. Our implementation is inspired by the training procedures described in Encodec (Défossez et al., 2022) and SpeechTokenizer (Zhang et al., 2024a). The RVQ projects input vectors to the most similar entry in a codebook, and the residual is calculated and processed in subsequent quantization steps, each utilizing a different codebook. The codebook entries are updated using an *exponential moving average* (EMA) with a *decay rate* of 0.99 for the matched item, while unmatched entries are replaced by candidates from the current batch. To ensure proper gradient flow during training, we employ a *straight-through estimator*. A *commitment loss* is also computed and added to the total training loss to promote stability. In our experiments, we utilize a codebook size of 1024 and 8 quantization levels.

**Discriminators.** We incorporate a trio of discriminators to enhance the quality and realism of the generated speech: the Multi-Scale Discriminator (MSD), the Multi-Period Discriminator (MPD), and the Multi-Scale Short-Time Fourier Transform (MS-STFT) discriminator. The MS-STFT discriminator follows the implementation outlined in (Défossez et al., 2022), operating on the real and imaginary components of multi-scale complex-valued STFTs. It begins with a 2D convolutional layer, followed by 2D convolutions with increasing dilation rates in the time dimension (1, 2, and 4) and a stride of 2 across the frequency axis in each sub-network. A final 2D convolution with a kernel size of 3 × 3 and a stride of (1, 1) is applied to produce the prediction. The MSD and MPD discriminators follow the architectures introduced in (Kong et al., 2020), with adjustments to the channel numbers to align the parameter count more closely with the MS-STFT discriminator. This ensemble of discriminators works in concert to provide comprehensive feedback on various aspects of the generated speech, contributing to the overall quality and naturalness of the output.

## E   RELATED WORK

**Tokenization Techniques in Speech.** Tokenization in speech processing can be broadly categorized into two main approaches: (i) speech encoder-based and (ii) language-based. In the speech encoder-based tokenization approach, a pretrained speech encoder serves as a teacher model, providing semantically rich audio representations. These representations are then used to guide the training model, either through an alignment network (Messica & Adi, 2024) or by optimizing specific losses (Zhang et al., 2024a; Liu et al., 2024). Language-based tokenization approach involves processing audio through a speech encoder to obtain discrete representations or using the corresponding text to feed into a language model. The representations from the language model are then utilized either to learn a tokenizer for speech (Turetzky & Adi, 2024) or to reconstruct speech (Hassid et al., 2024; Zhang et al., 2024b; Wang et al., 2024). Besides, (Zhang et al., 2024b) proposed SpeechLM where two discrete tokenizers were introduced and learned in an unsupervised way and converted the speech and text in a shared discrete space.

**Discrete Speech Representation.** There are two well-known methods for discrete speech representation: semantic tokens and acoustic tokens. Semantic tokens are derived through self-supervised learning (SSL) techniques for speech (Baevski et al., 2019; Hsu et al., 2021; Chung et al., 2021) and capture abstract, high-level features that relate to general, symbolic aspects of speech, while omitting details related to speaker identity and acoustic characteristics. In contrast, acoustic tokens are obtained using neural audio codecs (Zeghidour et al., 2021; Défossez et al., 2022; Yang et al., 2023) and focus on delivering precise reconstructions of acoustic features. However, recent models (Turetzky & Adi, 2024; Liu et al., 2024; Shi et al., 2024) have shown that speech models based on self-supervised learning (SSL) are effective at extracting acoustic representations where LMs be employed to refine these models further, enhancing their ability to extract more nuanced semantic representations.

**Textual Language Models in Speech.** Research on speech models, including works by (Nguyen et al., 2023), (Borsos et al., 2023), and (Kharitonov et al., 2022), has focused on utilizing raw audio to extract prosodic features, identify speaker characteristics, and generate audio without depending on textual features or supervision from textual LMs. In contrast, many newer methods have started using audio encoders to transform audio signals into discrete tokens, which can be processed by textual LMs. *TWIST* method introduced by (Hassid et al., 2024) initializes the weights of the SpeechLM using a pre-trained text LM, showing that this combination significantly improves performance. Similarly, the *SELM* model developed by (Wang et al., 2024) leverages GPT (Radford, 2018; Radford et al., 2019) as its foundation due to its enhanced parallel processing capabilities and capacity. However, text-based LLMs such as GPT-3 (Brown, 2020) and Llama (Touvron et al., 2023) are essential for speech modeling. Once discrete audio representations are obtained, these large text models are trained to align with or enhance the original text embedding space, as explored in studies by (Zhang et al., 2023), (Fathullah et al., 2023), (Shu et al., 2023), and (Rubenstein et al., 2023). This trend of integrating textual LMs into speech modeling has become increasingly popular in recent research.

## F   RECONSTRUCTED SPEECH COMPARISON

We plot the Mel-Spectrogram of the original speech and the reconstructed speech from DM-Codec and compare them with the reconstructed speech of EnCodec, SpeechTokenizer, and FACodec. Fine-grained differences may not be readily apparent in the Mel-Spectrogram visually; therefore, we encourage readers to click on the respective play button in Figure 4 for a hyperlink to the playable audio file.

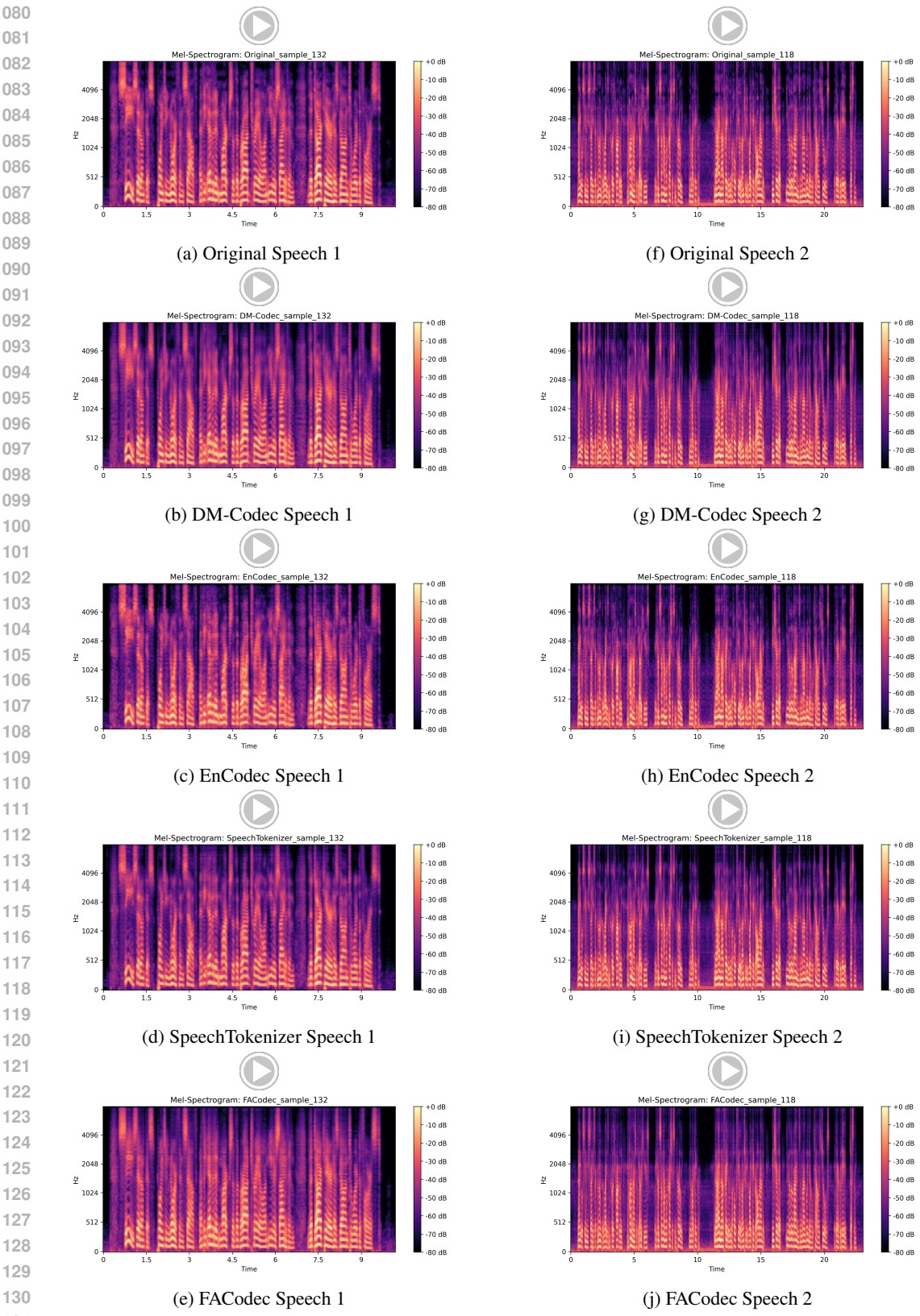

Figure 4: Reconstructed speech examples with clickable play buttons above each Mel-spectrogram.

## G   HUMAN EVALUATION METHODOLOGY

To evaluate the quality and effectiveness of our approach, we conducted human evaluations using Mean Opinion Score (MOS) and Similarity Mean Opinion Score (SMOS) metrics, following methodologies established in prior works such as SpeechTokenizer and Vall-E. The study was conducted under an approved Institutional Review Board (IRB) protocol to ensure ethical compliance and participant safety. A total of 50 proficient English speakers, comprising graduate and undergraduate students, were selected as evaluators based on their high language comprehensibility. These participants volunteered for the evaluation, were briefed on the study's purpose, and were provided no information that could bias their judgments.

The evaluation process involved each participant rating batches of fully anonymized and randomized speech samples via a web-based survey interface, with clear and standardized guidelines to ensure consistent and unbiased scoring. Each batch contained 16 samples, including outputs from both our proposed models and baseline systems. For the speech reconstruction task, participants rated the perceptual quality of the speech samples using the MOS, based on criteria such as naturalness, intelligibility, and clarity, employing a 5-point Likert scale, where higher scores indicated superior quality. For the text-to-speech evaluation, participants provided two distinct ratings: MOS, to measure the overall naturalness of the generated speech, and SMOS, to evaluate the similarity of the generated speech to the original speaker's voice, both rated on a 1-to-5 scale with 1-point increments. To enhance reliability and mitigate individual evaluator bias, each sample was rated by multiple participants.

**How natural, intelligible, and clear is the speech of each model?** *

**Google Drive Link with Sample Batches**

*Mean Opinion Score (MOS)*: Rate the quality of each sample based on how natural, intelligible, and clear the speech sounds.

**1**: Poor quality (unintelligible, very distorted)
**2**: Fair quality (hard to understand, but some parts are clear)
**3**: Good quality (understandable, but may have some distortions)
**4**: Very good quality (mostly clear, with minimal distortion)
**5**: Excellent quality (natural, clear, no distortions)

|              | 1 | 2 | 3 | 4 | 5 |
|--------------|---|---|---|---|---|
| _real_*      | ○ | ○ | ○ | ○ | ○ |
| model_A_fake_* | ○ | ○ | ○ | ○ | ○ |
| model_B_fake_* | ○ | ○ | ○ | ○ | ○ |
| model_C_fake_* | ○ | ○ | ○ | ○ | ○ |
| model_D_fake_* | ○ | ○ | ○ | ○ | ○ |
| model_E_fake_* | ○ | ○ | ○ | ○ | ○ |
| model_F_fake_* | ○ | ○ | ○ | ○ | ○ |
| model_G_fake_* | ○ | ○ | ○ | ○ | ○ |

(a) Interface View 1

Name ↑

🎧 _real_1089-134686-0000.flac 👥

🎧 model_A_fake_1089-134686-0000.flac 👥

🎧 model_B_fake_1089-134686-0000.flac 👥

🎧 model_C_fake_1089-134686-0000.flac 👥

🎧 model_D_fake_1089-134686-0000.flac 👥

🎧 model_E_fake_1089-134686-0000.flac 👥

🎧 model_F_fake_1089-134686-0000.flac 👥

🎧 model_G_fake_1089-134686-0000.flac 👥

(a) Interface View 2

**How natural is the synthesized speech?** *

**Google Drive Link with Sample Batches**

*Mean opinion score (MOS)*: Rate the naturalness of the speech.

**1**: Completely unnatural
**2**: Mostly unnatural
**3**: Neutral
**4**: Mostly natural
**5**: Completely natural

|  | 1 | 2 | 3 | 4 | 5 |
|---|---|---|---|---|---|
| 1_model_X_* | ○ | ○ | ○ | ○ | ○ |
| 1_model_Y_* | ○ | ○ | ○ | ○ | ○ |
| 1_model_Z_* | ○ | ○ | ○ | ○ | ○ |
| 2_model_X_* | ○ | ○ | ○ | ○ | ○ |
| 2_model_Y_* | ○ | ○ | ○ | ○ | ○ |
| 2_model_Z_* | ○ | ○ | ○ | ○ | ○ |

(a) Interface View 3

**How similar is the speech to the original speaker's voice?** *

**Google Drive Link with Sample Batches**

*Similarity mean opinion score (SMOS):* Rate how similar the speech is to the original speaker's voice.

**1**: Not similar at all
**2**: Slightly similar
**3**: Moderately similar
**4**: Very similar
**5**: Identical to the original speaker

|  | 1 | 2 | 3 | 4 | 5 |
|---|---|---|---|---|---|
| 1_model_X_* | ○ | ○ | ○ | ○ | ○ |
| 1_model_Y_* | ○ | ○ | ○ | ○ | ○ |
| 1_model_Z_* | ○ | ○ | ○ | ○ | ○ |
| 2_model_X_* | ○ | ○ | ○ | ○ | ○ |
| 2_model_Y_* | ○ | ○ | ○ | ○ | ○ |
| 2_model_Z_* | ○ | ○ | ○ | ○ | ○ |

(a) Interface View 4

Name ↑

1_1089_audio_prompt.flac

1_1089_text_input.txt

1_model_X_gen-1089-134691-0015.wav

1_model_Y_gen-1089-134691-0015.wav

1_model_Z_gen-1089-134691-0015.wav

2_model_X_gen-p225_056.wav

2_model_Y_gen-p225_056.wav

2_model_Z_gen-p225_056.wav

2_p225_audio_prompt.flac

2_p225_text_input.txt

(a) Interface View 5

Figure 9: Web-based survey interface and questionnaire used for evaluation.

## H  LIMITATIONS AND BROADER IMPACT

**Limitations.** In this work, we present the effectiveness of our proposed method, DM-Codec, based on the LibriSpeech dataset. Future research could investigate its performance across a variety of datasets and domains. Additionally, exploring the capabilities of DM-Codec in multilingual contexts would be valuable. Another limitation of our work is the absence of experiments with emerging LLMs. Currently, we focus solely on masked language models to derive representations. Further investigation into these decoder-based LLMs' impact on DM-Codec can be studied and addressed.

**Broader Impact.** The integration of language models in speech processing has traditionally focused on model-specific implementations or specific training objectives. In this work, we propose a novel approach by leveraging language models during the tokenization phase through our model, DM-Codec. By incorporating language-specific representations from the corresponding text, DM-Codec enhances the quality of discrete speech representations. This method bridges the gap between language and speech models, offering a more unified approach to multimodal representation learning. DM-Codec provides a robust framework for generating high-quality audio representations, with potential applications in various domains, including multilingual speech processing, low-resource languages, and other audio-related tasks. Our findings pave the way for more effective and contextually aware speech processing models, contributing to advancements in the broader field of speech and language technologies.

