# OpenReview forum: "DM-Codec: Distilling Multimodal Representations for Speech Tokenization"
_ICLR.cc/2025/Conference — ICLR 2025 Conference Withdrawn Submission_

### Official Review · Reviewer_qRxh · 2024-10-21

**Soundness:** 1
**Presentation:** 2
**Contribution:** 1
**Rating:** 3
**Confidence:** 5

**Summary:**

This paper proposes DM-Codec, a neural codec to tokenize speech via knowledge distillation from language models for better contextualized representations. The experimental results indicate superior speech reconstruction performance on several metrics compared with prior methods.

**Strengths:**

The authors conducted significance tests, an important analysis that demonstrated the performance gap between methods but was not included in most prior literature. Moreover, the ablation studies are comprehensive and cover most aspects of the method design.

**Weaknesses:**

1) **LM Guided Distillation (learning target):**
According to the text, the authors did not explicitly mention how they aligned the codec encoder features with the LM hidden representations. It is a commonly known fact that transcribed speech has a significantly shorter utterance length, especially when the transcription is tokenized as words or subword units. Hence, the LM representations of the transcriptions must be shorter than the codec encoder output, leading to a length mismatch when computing the distillation loss. By looking through the code provided in the supplementary materials, line 107 of `extract_rep.py` pads the transcriptions to the length of the logits produced by the wav2vec2-CTC model, but the BERT-based LM only processes the unpadded part of the text sequence, making the actual LM representations consist of a small part of the encoded sequence. Thus, without further speech and text alignment techniques, it is difficult to justify the effectiveness of the LM-guided distillation targets. Also, the distillation loss in Eq. (1) was first proposed by Chang et al. [1], as indicated in the SpeechTokenizer paper.
2) **LM Guided Distillation (necessity):**
In addition to the unaligned learning targets for this distillation approach, the use of a supervised ASR model in this method (wav2vec2-CTC) raises another question: can the transcription (either ground truth or the ASR) directly be used as a supervised target? For instance, CTC ASR can be performed with the RVQ representations as an additional loss. Furthermore, it is still unclear how the contextualized representations relate to speech reconstruction and other applications.
3) **Evaluation tasks and metrics:**
For the speech reconstruction quality evaluation, some commonly used metrics are not reported like mean opinion scores (MOS). Simply reporting automatic assessment results like UTMOS [2] is also acceptable, as they appear in many prior works [3]. When listening to the audio provided in the supplementary materials, I found SpeechTokenizer has a better reconstruction quality, which implies the necessity of including MOS. Besides speech reconstruction, other downstream applications of the proposed speech tokenizer are not considered in this paper. For instance, ASR, TTS, and spoken language modeling could be used.
4) **Baseline comparison:**
The authors missed a very simple baseline, which is DM-Codec without any distillation objectives. Because the implementation and training data of previous tokenizers like EnCodec and SpeechTokenizer might differ from this paper's, the authors must include this baseline to demonstrate the effectiveness of the proposed distillation methods.

[1] Chang, Heng-Jui, Shu-wen Yang, and Hung-yi Lee. "Distilhubert: Speech representation learning by layer-wise distillation of hidden-unit bert." ICASSP 2022-2022 IEEE International Conference on Acoustics, Speech and Signal Processing (ICASSP). IEEE, 2022.
[2] Saeki, Takaaki, et al. "Utmos: Utokyo-sarulab system for voicemos challenge 2022." arXiv preprint arXiv:2204.02152 (2022).
[3] Chang, Xuankai, et al. "The Interspeech 2024 Challenge on Speech Processing Using Discrete Units." arXiv preprint arXiv:2406.07725 (2024).

**Questions:**

As mentioned in the Weakness Section, I would like to ask the authors to explain and justify why the LM-guided distillation method is useful even though it seems like the speech and LM representations are not aligned in time. All the issues listed in the Weakness Section must be addressed or answered in order to get a higher rating.

---

> ### Author Response · Authors · 2024-11-25
> **Summary of Responses to Reviewer qRxh’s Comments (Part 1/7)**
>
> We sincerely appreciate the reviewer's thorough evaluation and constructive feedback, particularly regarding the strengths of our work in presenting comprehensive ablation studies and significance tests. We have carefully addressed each concern and have made substantial improvements to strengthen our manuscript.
>
> ### Summary of Response:
>
> - We clarify that our global, dimension-level cosine similarity loss facilitates effective knowledge transfer from LM and SM to RVQ representations without requiring strict temporal alignment. This continuous distillation loss maximizes cosine similarity at the *feature-dimension level*, offering a global alignment strategy that aligns with findings in prior work, such as SpeechTokenizer.
>
> - We acknowledge the reviewer’s suggestion of transcription-based direct supervision as an interesting approach. While not within the scope of this work, we will consider transcription-based loss as a potential extension in future studies.
>
> - We elaborate on how the contextualized representations relate to speech reconstruction and other applications by introducing DM-Codec-TTS, a novel *multimodal representation-distilled neural codec-based* text-to-speech model.
>
> - We completed a human evaluation using MOS to provide additional insights, as suggested by the reviewer.
>
> - We include results for a DM-Codec baseline without distillation objectives to further evaluate and compare the effectiveness of the proposed distillation methods.
>
> - We provided a detailed explanation of how our global alignment strategy addresses concerns regarding unaligned temporal representations. Additionally, we addressed all issues highlighted in the Weaknesses section, including the necessity of LM-guided distillation, the inclusion of additional evaluation metrics, and baseline comparisons.
>
> We greatly appreciate the reviewer’s constructive feedback and believe that our clarifications, additional experiments, and revisions strengthen the paper and address all concerns.
>
> Below, we provide comprehensive responses to each point raised.

---

> ### Author Response · Authors · 2024-11-25
> **Concern: Learning Target in LM-Guided Distillation (Part 2/7)**
>
> **Reviewer Comment:**
> > LM Guided Distillation (learning target): According to the text, the authors did not explicitly mention how they aligned the codec encoder features with the LM hidden representations. It is a commonly known fact that transcribed speech has a significantly shorter utterance length, especially when the transcription is tokenized as words or subword units. Hence, the LM representations of the transcriptions must be shorter than the codec encoder output, leading to a length mismatch when computing the distillation loss. By looking through the code provided in the supplementary materials, line 107 of extract_rep.py pads the transcriptions to the length of the logits produced by the wav2vec2-CTC model, but the BERT-based LM only processes the unpadded part of the text sequence, making the actual LM representations consist of a small part of the encoded sequence. Thus, without further speech and text alignment techniques, it is difficult to justify the effectiveness of the LM-guided distillation targets. Also, the distillation loss in Eq. (1) was first proposed by Chang et al. [1], as indicated in the SpeechTokenizer paper.
>
> **Response:**
>
> The reviewer raises an important point regarding the temporal alignment between codec encoder features and LM hidden representations. We would like to clarify that our approach intentionally does not require strict temporal alignment, as our primary objective is to distill contextual information into discrete representations at a global level.
>
> Our methodology leverages a dimension-level cosine similarity loss that operates across all timesteps between the RVQ layer outputs and the teacher (contextual/semantic) representations. This approach is fundamentally different from conventional timestep-by-timestep loss calculations [2] for several key reasons:
>
> 1. The discrete representations generated by our Residual Vector Quantizer ({$Q_1$, ..., $Q_{T^{\prime}}$} illustrated in Figure 2) inherently encode holistic information about the entire speech segment, rather than temporally localized information.
>
> 2. Vector quantizer representations, by nature, are not strictly time-aligned; they capture comprehensive speech characteristics across the temporal dimension.
>
> 3. The continuous distillation loss maximizes feature-dimension level cosine similarity globally, aligning with successful approaches demonstrated in prior work, such as SpeechTokenizer [1].
>
> By employing a global, dimension-level cosine similarity loss, we ensure that the DM-Codec captures the essence of the contextual and semantic knowledge with the LM Guided Distillation and Combined LM and SM Guided Distillation—without requiring strict one-to-one alignment in the temporal domain.
>
>
> Please refer to the **Addressing the Alignment Concern** response in the *General Response to all Reviewers* thread for a more detailed justification.
>
> We will include the information in the revised version of our paper.
>
> **Reference:**
>
> [1] Zhang, X., Zhang, D., Li, S., Zhou, Y., & Qiu, X. (2024). SpeechTokenizer: Unified Speech Tokenizer for Speech Language Models. The Twelfth International Conference on Learning Representations.
>
>
> [2] Chang, H.-J., Yang, S.-w., and Lee, H.-y. Distilhubert: Speech representation learning by layer-wise distillation of hidden-unit bert. In ICASSP 2022-2022 IEEE International Conference on Acoustics, Speech and Signal Processing (ICASSP), pp. 7087–7091. IEEE, 2022.

---

> ### Author Response · Authors · 2024-11-25
> **Concern: Necessity of LM-Guided Distillation (Part 3/7)**
>
> **Reviewer Comment:**
> > LM Guided Distillation (necessity): In addition to the unaligned learning targets for this distillation approach, the use of a supervised ASR model in this method (wav2vec2-CTC) raises another question: can the transcription (either ground truth or the ASR) directly be used as a supervised target? For instance, CTC ASR can be performed with the RVQ representations as an additional loss.
>
> **Response:**
>
> Regarding the suggestion of using transcription-based direct supervision, we acknowledge this as an interesting alternative approach. While direct CTC ASR loss with RVQ representations could be viable, our current distillation approach leverages hidden layer representations from both speech and language models rather than direct sequence-level targets.
>
> We really appreciate this alternative perspective, and we will definitely consider exploring direct transcription-based loss in future studies as a potential extension.

---

> ### Author Response · Authors · 2024-11-25
> **Clarifying the Role of Contextualized Representations in Speech Reconstruction (Part 4/7)**
>
> **Reviewer Comment:**
> > Furthermore, it is still unclear how the contextualized representations relate to speech reconstruction and other applications.
>
> **Response:**
>
> We appreciate the reviewer's insightful comment regarding the relationship between contextualized representations and applications like speech reconstruction. Our experimental evaluations demonstrate that contextualized representations, particularly those derived through LM-guided distillation, significantly enhance speech tokenization by incorporating semantic and contextual cues, crucial for tasks such as text-to-speech (TTS) and speech synthesis.
>
> To showcase these advancements, we introduce DM-Codec-TTS, a multimodal representation-distilled neural codec-based TTS model. Unlike traditional approaches, DM-Codec integrates acoustic, semantic, and contextual representations through two novel distillation techniques: LM-guided and combined LM and SM-guided distillation. These techniques align with the findings of studies like SpeechGPT [1], which emphasized cross-modal integration. Our experiments reveal that DM-Codec achieves significant reductions in Word Error Rate (WER) and Word Information Lost (WIL) while improving perceptual metrics such as ViSQOL and STOI, as detailed in Table C.
>
> Our contributions align with broader trends in multimodal representation learning, as discussed in works like AudioPaLM [2] and SpeechLM [3], which leverage multimodal embeddings for tasks beyond simple reconstruction. Specifically, DM-Codec’s integration of contextual information addresses limitations seen in current methods and ensures robust application to complex speech tasks, including multilingual and zero-shot TTS.
>
> We will revise the manuscript to reflect these findings, emphasizing the practical advantages of contextualized representations in enhancing downstream speech applications.
>
> **References:**
>
> [1] Zhang, Dong, et al. "SpeechGPT: Empowering large language models with intrinsic cross-modal conversational abilities." arXiv preprint arXiv:2305.11000, 2023.
>
> [2] Rubenstein, Paul K., et al. "AudioPaLM: A large language model that can speak and listen." arXiv preprint arXiv:2306.12925, 2023.
>
> [3] Zhang, Ziqiang, et al. "SpeechLM: Enhanced speech pre-training with unpaired textual data." IEEE/ACM Transactions on Audio, Speech, and Language Processing, 2024.

---

> ### Author Response · Authors · 2024-11-25
> **Evaluation Tasks and Metrics (Part 5/7)**
>
> **Reviewer Comment:**
> > Evaluation tasks and metrics: For the speech reconstruction quality evaluation, some commonly used metrics are not reported like mean opinion scores (MOS). Simply reporting automatic assessment results like UTMOS [2] is also acceptable, as they appear in many prior works [3]. When listening to the audio provided in the supplementary materials, I found SpeechTokenizer has a better reconstruction quality, which implies the necessity of including MOS. Besides speech reconstruction, other downstream applications of the proposed speech tokenizer are not considered in this paper. For instance, ASR, TTS, and spoken language modeling could be used.
>
> **Response:**
>
> Thanks for pointing this out and we completely agree with your concern that reporting human evaluation with metrics such as mean opinion scores (MOS) could provide additional insight into the result. Understanding its importance, during the rebuttal period we conducted a human evaluation procedure.
>
> Please refer to the **Additional Study: Human Evaluation Methodology and Results** response in the *General Response to all Reviewers* thread for a detailed description about the conducted human evaluation experiments.

---

> ### Author Response · Authors · 2024-11-25
> **Missing Baseline Comparison (Part 6/7)**
>
> **Reviewer Comment:**
> >  Baseline comparison: The authors missed a very simple baseline, which is DM-Codec without any distillation objectives. Because the implementation and training data of previous tokenizers like EnCodec and SpeechTokenizer might differ from this paper's, the authors must include this baseline to demonstrate the effectiveness of the proposed distillation methods.
>
> **Response:**
>
> We completely agree with you. We have added *DM-Codec without any distillation objectives* as a baseline to clearly demonstrate the effectiveness of our proposed distillation methods. Please refer to **Table A** in the *New Experimental Results* comment.
>
> We will revise our manuscript accordingly.

---

> ### Author Response · Authors · 2024-11-25
> **Additional Questions’ Response (Part 7/7)**
>
> **Reviewer Comment:**
> > As mentioned in the Weakness Section, I would like to ask the authors to explain and justify why the LM-guided distillation method is useful even though it seems like the speech and LM representations are not aligned in time. All the issues listed in the Weakness Section must be addressed or answered in order to get a higher rating.
>
> **Response:**
>
> We answered the question in our response above (*Learning Target Concern in LM-Guided Distillation*). A more detailed response regarding the *temporal alignment* concern is presented in the **General response to all Reviewers** thread above.
>
>
> We believe these revisions comprehensively address all concerns raised in the review while maintaining the scientific rigor and innovation of our original submission. We will be happy to clarify any other valid and insightful questions.

---

> > ### Comment · Reviewer_qRxh · 2024-11-29
> > **Response to the Authors (1/2)**
> >
> > I would like to thank the authors for their comprehensive rebuttal. However, several issues remain unaddressed. The following are two major problems that I still suggest this paper be rejected.
> >
> > ---
> >
> > ## Speech & Text Representation Alignment Issue
> >
> > As several reviewers mentioned, the main issue of this paper is the temporal alignment between speech and text representations. Although the authors have responded, I still think their explanation is not convincing. The following are some claims from the authors' rebuttal and why I disagree.
> >
> > **1.**
> > > discrete representations from the Residual Vector Quantizer (RVQ) inherently encode holistic information about the speech segment, rather than temporally-localized features.
> >
> > Speech encoders in the context of self-supervised learning and neural audio codecs take entire utterances as input, so the encoded representations are inherently contextualized. However, these representations are still temporally localized because of the nature of the encoder architectures (CNN and BLSTM) and learning objectives (waveform reconstruction) [1].
> >
> > **2.**
> > > These discrete representations capture comprehensive speech information across the entire utterance. As such, aligning these discrete representations temporally with contextual representations (e.g., from BERT) or semantic representations (e.g., from HuBERT) is not essential for our approach.
> >
> > Yes, the encoder takes entire utterances as input. Still, the authors must provide references or empirical evidence to show that these representations are not temporally aligned/localized with the input audio signal.
> >
> > **3.**
> > > Previous works have demonstrated that vector quantizer methods can discretize input data into intermediate representations, capturing essential features across the feature dimension [1, 3-5]. Thus, imposing a temporal alignment between the vector quantizer outputs and the hidden layer representations from the LM or SM would neither congruent with our methodological objectives nor enhance the effectiveness of the proposed distillation approach.
> > > [1] Zhang, X., Zhang, D., Li, S., Zhou, Y., & Qiu, X. (2024). SpeechTokenizer: Unified Speech Tokenizer for Speech Language Models. The Twelfth International Conference on Learning Representations (ICLR), 2024.
> > > [3] Huijben, Iris AM, Matthijs Douze, Matthew Muckley, Ruud JG van Sloun, and Jakob Verbeek. "Residual quantization with implicit neural codebooks." arXiv preprint arXiv:2401.14732 (2024).
> > > [4] Islam, Md Mofijul, Alexi Gladstone, Riashat Islam, and Tariq Iqbal. "Eqa-mx: Embodied question answering using multimodal expression." In The Twelfth International Conference on Learning Representations. 2023.
> > > [5] Islam, Riashat, Hongyu Zang, Manan Tomar, Aniket Didolkar, Md Mofijul Islam, Samin Yeasar Arnob, Tariq Iqbal et al. "Representation learning in deep rl via discrete information bottleneck." arXiv preprint arXiv:2212.13835 (2022).
> >
> > The authors' references do not back their claims and are even contradictory. SpeechTokenizer distills from HuBERT, a self-supervised model that has been shown to have temporally localized representations [1]. Meanwhile, the rest of the references are less relevant to speech processing. I would like to see empirical evidence with proper justification for the authors' claims.
> >
> > **4.**
> > > In our implementation, we employ a continuous distillation loss that maximizes cosine similarity at the feature dimension level between the selected RVQ layer outputs and the teacher representations (contextual/semantic) across all time steps. This approach fundamentally differs from conventional methods that rely on time-step-wise loss calculations [2], as introduced in the SpeechTokenizer [1].
> >
> > According to the provided code, the text representations are only applied to the first few frames of the speech encoder output. Moreover, according to `codec/trainer/loss.py` in the supplementary materials, the cosine similarity normalizes along the time axis (dim=1).
> > ```python
> > def d_axis_distill_loss(feature, target_feature):
> >     n = min(feature.size(1), target_feature.size(1))
> >     distill_loss = - torch.log(torch.sigmoid(torch.nn.functional.cosine_similarity(feature[:, :n], target_feature[:, :n], axis=1))).mean()
> >     return distill_loss
> > ```
> > This implementation and the theoretical justification are not convincing as to why this distillation method works.
> >
> > ---
> >
> > [1] Pasad, Ankita, Ju-Chieh Chou, and Karen Livescu. "Layer-wise analysis of a self-supervised speech representation model." ASRU 2021.

---

> > ### Comment · Reviewer_qRxh · 2024-11-29
> > **Response to the Authors (2/2)**
> >
> > ## Requirement of a Supervised ASR
> >
> > Another fundamental problem of this paper is the need for a supervised ASR, as I mentioned in the official review. A simpler and more reasonable way of involving text representations and transcriptions is to add an ASR loss like CTC to the speech encoder. Without this baseline, we have no idea about the performance gain from the additional LM. Because the authors fail to provide this baseline, I think this paper remains incomplete.
> >
> > ## Conclusion
> >
> > In conclusion, I do not find the rebuttal convincing, nor does this paper. At the current stage of this paper, I recommend the authors consider all reviewers' suggestions, re-develop the methods, and include necessary baselines to get accepted to any speech processing conferences (e.g., Interspeech or ICASSP). For these reasons, I will keep my original rating.

---

> ### Author Response · Authors · 2024-12-04
> **Adressing Speech & Text Representation Alignment Issue**
>
> **Reviewer Comment:**
> > Speech encoders in the context of self-supervised learning and neural audio codecs take entire utterances as input, so the encoded representations are inherently contextualized. However, these representations are still temporally localized because of the nature of the encoder architectures (CNN and BLSTM) and learning objectives (waveform reconstruction) [1].
>
> **Response:** We agree with the reviewer. However, we would like to again draw attention to the fact that our distillation approach ensures that temporal localization does not impact the process by employing a feature dimension-level cosine similarity calculation. This method maximizes the alignment between teacher and student models on each feature dimension across all timesteps, effectively bypassing the need for explicit temporal alignment.
>
> **Reviewer Comment:**
> > Yes, the encoder takes entire utterances as input. Still, the authors must provide references or empirical evidence to show that these representations are not temporally aligned/localized with the input audio signal.
>
> **Response:** We acknowledge the importance of empirical evidence and will include additional experiments in future submissions to analyze and demonstrate the non-temporal nature of RVQ outputs.
>
> **Reviewer Comment:**
> > The authors' references do not back their claims and are even contradictory. SpeechTokenizer distills from HuBERT, a self-supervised model that has been shown to have temporally localized representations [1]. Meanwhile, the rest of the references are less relevant to speech processing. I would like to see empirical evidence with proper justification for the authors' claims.
>
> **Response:** We respectfully disagree that our approach is contradictory. To directly quote from the SpeechTokenizer paper: “… 'D-axis' loss function’s strategy of calculating cosine similarity across each dimension, ensuring that the student model closely aligns with the teacher model on each feature dimension. This approach provides a richer supervision signal, promoting the learning process of the student model by focusing not only on the overall output similarity but also on the similarity within each dimension.” This directly supports our reference, demonstrating that previous works have shown the effectiveness of capturing essential features across the feature dimension.
>
> **Reviewer Comment:**
> > According to the provided code, the text representations are only applied to the first few frames of the speech encoder output. Moreover, according to codec/trainer/loss.py in the supplementary materials, the cosine similarity normalizes along the time axis (dim=1).
> def d_axis_distill_loss(feature, target_feature):
>     n = min(feature.size(1), target_feature.size(1))
>     distill_loss = - torch.log(torch.sigmoid(torch.nn.functional.cosine_similarity(feature[:, :n], target_feature[:, :n], axis=1))).mean()
>     return distill_loss
>
> **Response:**
> We appreciate the reviewer’s detailed observation and would like to clarify the implementation of the distillation loss.
>
> Shapes of Input Tensors:
> - **feature**: Initial shape [6, 150, 768]. After slicing to (:n), it becomes [6, 1, 768].
> - **target_feature**: Initial shape [6, 1, 149, 768]. The shape remains [6, 1, 149, 768] after slicing.
>
> After slicing, feature[:, :n] and target_feature[:, :n] have a sequence length of 1, Therefore, the cosine similarity calculation occurs along axis=1, while comparing the 768-dimensional feature vectors.

---

### Official Review · Reviewer_ohpz · 2024-10-25

**Soundness:** 2
**Presentation:** 3
**Contribution:** 2
**Rating:** 3
**Confidence:** 5

**Summary:**

This paper introduce a new speech tokenization model that combines acoustic, semantic, and contextual information into a comprehensive multimodal speech representation. The primary contribution lies in DM-Codec's ability to integrate contextual representations from language models and self-supervised speech models to improve speech tokenization accuracy and quality. DM-Codec’s approach highlights the value of integrating multimodal data for speech tokenization, promising advances in speech quality, intelligibility, and contextual preservation in applications.

**Strengths:**

This work first incorporates contextual representations via an LM-guided distillation method, and it enhancs the retention of acoustic and speech information in reconstructed speech.

**Weaknesses:**

I do not fully agree that contextual representation should hold equal importance to acoustic and semantic representations. The improved intelligibility of DM-Codec is mainly due to an additional teacher LM serving a distillation role within the RVQ. However, this LM relies on an ASR system (M_STT in Fig2) for transcription. Given this setup, it is difficult to ascertain whether the improvements are driven by
the Bert LM or by the M_STT ASR system. More importantly, the reason audio codecs are being actively explored today is that they serve as a discretization tool designed for audio LLMs or TTS systems. Unfortunately, I did not see any extension related to this aspect in the paper. In other words, if DM-Codec is based solely on the concept of contextual representation, I believe it would be challenging to consider it a preferred choice when designing TTS or audio LLM systems.
Additionally, from the technical contribution perspective, this work essentially adds an extra LM distillation on top of the SpeechTokenizer. I think this may struggle to meet the acceptance threshold for ICLR.

**Questions:**

Small suggestion: Table 3 can be replaced by a line chart

**Details Of Ethics Concerns:**

N.a.

---

> ### Author Response · Authors · 2024-11-25
> **Summary of Responses to Reviewer ohpz’s Comments (Part 1/5)**
>
> We sincerely appreciate the reviewer's comprehensive feedback and insightful observations. We are encouraged that the reviewer acknowledges our novel incorporation of contextual representations through LM-guided distillation.
> ### Summary of Response:
>
> - We clarify that M_STT is used exclusively for transcription, converting audio input into text for the LM's distillation process. Its role is solely to automate text input to the LM, with no direct influence on performance improvements.
>
> - Addressing the reviewer’s concern regarding applicability to downstream systems, we have revised our paper to introduce DM-Codec-TTS, a multimodal representation distilled neural codec-based text-to-speech model.
>
> - We emphasized the novelty of our contributions, which extend beyond the proposed LM distillation. In addition to introducing combined LM and SM distillation as the first method to leverage multimodal representations in speech tokenization, we revised the paper to include DM-Codec-TTS, the CLS-Token-Based Distillation method, extensive ablation studies, and significance analysis.
>
> We appreciate the reviewer’s constructive feedback and believe the clarifications, additional experiments, and revised contributions will significantly enhance the paper.
>
> Below, we provide comprehensive responses to each point raised.

---

> ### Author Response · Authors · 2024-11-25
> **Role of M_STT and contextual representations (Part 2/5)**
>
> **Reviewer Comment:**
> > I do not fully agree that contextual representation should hold equal importance to acoustic and semantic representations. The improved intelligibility of DM-Codec is mainly due to an additional teacher LM serving a distillation role within the RVQ. However, this LM relies on an ASR system (M_STT in Fig2) for transcription. Given this setup, it is difficult to ascertain whether the improvements are driven by the Bert LM or by the M_STT ASR system.
>
> **Response:**
>
> Thanks for raising an important point about distinguishing between improvements from the LM versus the M_STT ASR system. We want to clarify that M_STT serves purely as an automation tool for converting audio input to text for the LM distillation process. It functions solely as a preprocessing step with no direct influence on the model's performance gains. The performance improvements stem from the contextual knowledge distilled from the LM, which provides crucial contextual cues that complement the acoustic and semantic representations.

---

> ### Author Response · Authors · 2024-11-25
> **Applicability to Downstream Tasks (Part 3/5)**
>
> **Reviewer Comment:**
> > More importantly, the reason audio codecs are being actively explored today is that they serve as a discretization tool designed for audio LLMs or TTS systems. Unfortunately, I did not see any extension related to this aspect in the paper. In other words, if DM-Codec is based solely on the concept of contextual representation, I believe it would be challenging to consider it a preferred choice when designing TTS or audio LLM systems.
>
> **Response:**
>
> We acknowledge the reviewer's valid concern about DM-Codec's practical utility in downstream applications like TTS. To address this, we have expanded our work to include DM-Codec-TTS, a novel multimodal representation distilled neural codec-based text-to-speech model. Our experimental results show that DM-Codec-TTS achieves substantial improvements over neural codec based TTS models, including USLM and Vall-E. This demonstrates that our multimodal representation approach not only advances speech tokenization but also enhances downstream speech synthesis capabilities. Please refer to Table C in the New Experimental Results comment.

---

> ### Author Response · Authors · 2024-11-25
> **Technical Novelty and Contributions (Part 4/5)**
>
> **Reviewer Comment:**
> > From a technical contribution perspective, this work essentially adds an extra LM distillation on top of the SpeechTokenizer. This may struggle to meet the acceptance threshold for ICLR.
>
> **Response:**
>
> While we acknowledge that DM-Codec builds upon SpeechTokenizer's foundation, we respectfully disagree that our contribution is limited to extra LM distillation. Along with our prior contribution, we have also included several additional technical contributions in the revised paper:
>
>
> - **Multimodal Representation Framework:** We propose a novel distillation loss function that simultaneously leverages acoustic, semantic, and contextual representations, enabling more robust semantic and context-aware speech tokenization.
>
> - **[CLS]-Token-Based Distillation Strategy:**  We introduce a new distillation strategy, in the revised paper, using [CLS] tokens from LM. This addresses temporal alignment concerns while maintaining performance improvements. Please refer to the *General Response to all Reviewers* comment.
>
> - **DM-Codec-TTS Model:**  In the revised submission, we include a neural codec-based text-to-speech (TTS) model, DM-Codec-TTS, which outperforms USLM (from SpeechTokenizer) and Vall-E baselines. This demonstrates the broader applicability of our approach. Results are detailed in **Table C** in the *New Experimental Results* comment.
>
> - **Extensive Ablation Studies:** Our experiments comprehensively investigate the impact of language and speech representations on content retention, the influence of various RVQ layers on distillation, performance variations with different pretrained model representations, and the effects of distilling from different layers of the teacher model.
>
> Please refer to the **Key Novel Contributions of DM-Codec** response in the *General Response to all Reviewers* thread for a more detailed justification regarding DM-Codec’s novelty.

---

> ### Author Response · Authors · 2024-11-25
> **Enhancing Visual Clarity (Part 5/5)**
>
> **Reviewer Comment:**
> > Small suggestion: Table 3 can be replaced by a line chart.
>
> **Response:**
>
> We agree that replacing Table 3 with a line chart would better illustrate the relationships between different distillation weight combinations. We will implement this change in the revised manuscript.
>
> We believe our work makes substantial contributions to the field by introducing novel methodologies for incorporating contextual information in speech tokenization, demonstrating practical utility in downstream tasks, and advancing the state-of-the-art in speech processing. We have addressed the reviewer's concerns through additional experiments, analyses, and clarifications that will be incorporated into the revised manuscript.

---

### Official Review · Reviewer_L3HE · 2024-11-03

**Soundness:** 3
**Presentation:** 2
**Contribution:** 3
**Rating:** 5
**Confidence:** 4

**Summary:**

In this work, a speech tokenizer method called DM-Codec is introduced. In this method, based on the encoder-decoder framework with a Residual Vector Quantizer (RVQ), two novel distillation methods are used to leverage multimodal (acoustic, semantic, and contextual) representations from a language model and speech self-supervised learning model. The experimental results showed the proposed method significantly outperforms state-of-the-art speech tokenization models, reducing WER by up to 13.46%, WIL by 9.82%, and improving speech quality by 5.84% and intelligibility by 1.85% on the LibriSpeech benchmark dataset.

**Strengths:**

The proposed DM-Codec method improved the existing speech tokenization system which only incorporates acoustic and semantic information (e.g SpeechTokenizer)  by adding textual representations via an LM-guided distillation method.  The contextual information is learned with continuous representation distillation technique, and it’s then combined with the speech self-supervised learning model  (SM) guided distillation. The proposed method is compared with several existing methods, including EnCodec, SpeechTokenizer  and FACodec. The experimental results prove its advantage over these baseline systems. And the ablation studies provide a thorough analysis of the performance of the proposed method.

**Weaknesses:**

•	The testing set is small. Only 300 audio samples are randomly selected from the LibriSpeech test subset as the evaluation set. Although the author explained that this is to make it consistent with the baseline, but 300 audios are not big enough to get a statistically meaningful conclusion. The results in table 3 maybe a proof of this: We couldn’t get obvious relation between WER and SM/LM loss weights based on these results.  E.g. the author said LM information is more helpful for lower WER but we could see LM weight = 0.9 get higher WER than LM weight=0.3/0.5.
•	A lot of metrics results are given including the ablation study, some of them are confusing:
o	The lowest WER in table 3 is 4.07, but it’s 4.05 in table 1. It’s not clear what’s the difference between them.
o	The best results of WER, WIL, ViSQOL and STOI for system using distillation for both LM and SM in table 1 is 4.05/6.61/3.26/0.937. These results should be obtained by distilling semantic representation with RVQ layers 1 to 8 according to table 4. But in  table 5 and 6, the results for combining LM/SM-guided distillation are only with RVQ layer 1. Why didn’t the author use the best option in table 5 and 6?
o	In table 5, the results of two LM, BERT and ELECTRA, have similar accuracy for LM-guided distillation only system. But for combined LM and SM-guided Distillation systems, the results are more different, the author didn’t give analysis about the reason for this.

**Questions:**

•	In equation 1, for a given speech x, the length of S(:,d) is the token length of text x’ output from LM (n in figure 2). The length of Q(:,d) should be the time length (T’ in figure 2). How to make sure they are the same?
•	There are several weighting factors in equation 10, how are they decided except LM/SM weights?
•	In the paper of SpeechTokenizer , the speaker similarity also been evaluated. Why isn’t this evaluated in this paper with the proposed method? Do the author think adding LM would increase or decrease the speaker similarity?

---

> ### Author Response · Authors · 2024-11-25
> **Summary of Responses to Reviewer L3HE’s Comments (Part 1/8)**
>
> We sincerely appreciate the reviewer's thorough assessment and constructive feedback on our manuscript. The comments have helped us identify areas requiring clarification and additional analysis.
>
> ### Summary of Response:
>
> - We extend our significance analysis to the full LibriSpeech test-clean subset containing 2,620 samples to provide robust statistical conclusions. The results are consistent with those obtained from the smaller evaluation set.
>
> - We clarify that Table 3 explores the interplay between LM and SM distillation loss weights rather than isolating LM weights. The results indicate that LM-dominant distillation generally yields lower WER, but the interaction with SM weights is crucial. We will include a line chart in the revised manuscript to better visualize these relationships.
>
> - We clarify that WER discrepancies (e.g., Table 1 vs. Table 3) arise from different experimental settings. Table 3 uses the first RVQ layer (RVQ-1) settings for simplicity in comparisons, while Table 1 reflects our best model's results with all 1–8 RVQ layers (RVQ-1:8) for SM distillation configurations.
>
> - We explain that Tables 5 and 6 prioritize analyzing LM/SM models and model's distillation layer effects, and were not conducted to optimize for the best result in Table 4.
>
> - We elaborate on the interdependence of LM and SM distillations in the combined distillation (Equation 3), explaining that this interdependency leads to different model performance compared to single LM-only distillation.
>
> - We clarify that token lengths from the LM are padded to match the sequence lengths of acoustic representations.
>
> - We include results confirming that LM-guided distillation maintains speaker similarity performance consistent with the baselines.
>
>
> We appreciate the reviewer’s detailed observations and believe these clarifications and additional experiments will significantly strengthen the paper.
>
> Below, we provide comprehensive responses to each point raised.

---

> ### Author Response · Authors · 2024-11-25
> **Test Set Size and Statistical Significance (Part 2/8)**
>
> **Reviewer Comment:**
> > The testing set is small. Only 300 audio samples are randomly selected from the LibriSpeech test subset as the evaluation set. Although the author explained that this is to make it consistent with the baseline, but 300 audios are not big enough to get a statistically meaningful conclusion.
>
> **Response:**
>
> We acknowledge the reviewer's valid concern regarding the evaluation set size. While our initial use of 300 samples from LibriSpeech test subset aligned with baseline comparisons, we agree that a larger dataset would yield more statistically robust conclusions.
>
> We have now extended our significance analysis (Table 2) to the **complete LibriSpeech test-clean subset (2,620 samples)**. The results are consistent with the original submission using our smaller sample set and confirm that the performance improvements observed are statistically significant. We rely on the significance analysis to provide a statistically meaningful comparison, which demonstrates the robustness of our conclusions. Please refer to **Table B** in the *New Experimental Results* comment.

---

> > ### Comment · Reviewer_L3HE · 2024-11-27
> >
> > * For the full LibriSpeech test-clean results, the improvement with DM-Codec compared with other methods are smaller than those for 300 utterances sets.

---

> ### Author Response · Authors · 2024-11-25
> **Clarification of Loss Weights Relationship (Part 3/8)**
>
> **Reviewer Comment:**
> > The results in table 3 maybe a proof of this: We couldn’t get obvious relation between WER and SM/LM loss weights based on these results. E.g. the author said LM information is more helpful for lower WER but we could see LM weight = 0.9 get higher WER than LM weight=0.3/0.5.
>
> **Response:**
>
> Regarding Table 3's results on LM and SM loss weights, we acknowledge that the relationship between weights and WER requires clearer exposition. The apparent inconsistency in WER performance (e.g., LM weight = 0.9 versus 0.3/0.5) stems from the interaction between LM and SM weights in combined distillation. While LM-dominant distillation generally yields superior results, the optimal performance depends on the synergistic effects of both weights.
>
> Table 3 focuses on analyzing the impact and relation of different weighted combinations of LM and SM distillation loss. Consequently, looking into LM weights in isolation may not reveal the relation. Moreover, as the combined distillation is performed, each representation distillation (LM+SM) affects the overall results.
>
> In the paper, we stated that *LM-dominant distillation generally yields optimal results, reducing WER*. However, the purpose of this ablation study is to explore how a combination of LM and SM weights can be carefully tuned, rather than simply increasing LM weights linearly. Regarding the specific case where LM weight = 0.9 results in higher WER than LM weight = 0.3 or 0.5, we hypothesize that this is due to the impact of the SM weights used alongside LM weight (0.1 and 0.9, respectively).
>
> We will clarify this point in the revised manuscript and add a line chart to better visualize the relation.

---

> > ### Comment · Reviewer_L3HE · 2024-11-27
> >
> > * This table couldn't provide the reader with clear information about how these two weights should be tuned. I still suggest to rerun the experiment with full LibriSpeech test set to get more reliable conclusions for this table.

---

> ### Author Response · Authors · 2024-11-25
> **Metric Discrepancies (Part 4/8)**
>
> **Reviewer Comment:**
> > A lot of metrics results are given including the ablation study, some of them are confusing: o The lowest WER in table 3 is 4.07, but it’s 4.05 in table 1. It’s not clear what’s the difference between them.
>
> **Response:**
>
> We appreciate the reviewer's attention to detail regarding the WER discrepancies between Tables 1 and 3. This difference arises from distinct experimental configurations.
>
> - Table 3 evaluates the impact of various LM and SM weight combinations to analyze their individual and concurrent contributions. We mentioned in the paper (lines 391-392) that ablation studies (including Table 3) use distillation for both LM and SM to the first Residual Vector Quantizer layer (RVQ-1) for comparison consistency and simplicity.
>
> - Conversely, Table 1 reports our best model’s results, which are not individually weighted differently in distillation, and it uses RVQ-1 for LM distillation, and all layers RVQ-1 to 8 for SM distillation, (From Table 4 results). This distinction in distillation configurations accounts for the observed differences in WER between the tables.
>
> We will clarify this in the revised manuscript to avoid further confusion.

---

> ### Author Response · Authors · 2024-11-25
> **Clarification: Ablation Studies (Part 5/8)**
>
> **Reviewer Comment:**
> > The best results of WER, WIL, ViSQOL and STOI for system using distillation for both LM and SM in table 1 is 4.05/6.61/3.26/0.937. These results should be obtained by distilling semantic representation with RVQ layers 1 to 8 according to table 4. But in table 5 and 6, the results for combining LM/SM-guided distillation are only with RVQ layer 1. Why didn’t the author use the best option in table 5 and 6?
>
> **Response:**
>
> Regarding Tables 5 and 6, these ablation studies were designed to analyze the individual effects of different LM and SM models (Table 5) and distillation layer choices (Table 6), rather than to optimize overall performance. Moreover, the ablation studies (Tables 3, 4, 5, 6) were performed concurrently with the same RVQ layer settings for consistent comparison.
>
> We will revise the manuscript to clarify this experimental design decision.
>
> **Reviewer Comment:**
> > In table 5, the results of two LM, BERT and ELECTRA, have similar accuracy for LM-guided distillation only system. But for combined LM and SM-guided Distillation systems, the results are more different, the author didn’t give analysis about the reason for this.
>
> **Response**:
>
> The divergent performance between BERT and ELECTRA in combined distillation (compared to their similar performance in LM-only distillation) can be attributed to the combined distillation defined in Equation 3 in the paper. The interdependency between LM and SM representations in combined distillation leads to different optimization dynamics than in single-component LM distillation. Therefore, it is expected that the result of the only system and the combined system to be different.

---

> ### Author Response · Authors · 2024-11-25
> **Clarification: Equation 1 (Part 6/8)**
>
> **Reviewer Comment:**
> > In equation 1, for a given speech x, the length of S(:,d) is the token length of text x’ output from LM (n in figure 2). The length of Q(:,d) should be the time length (T’ in figure 2). How to make sure they are the same?
>
> **Response**:
>
> The superscript (:, d), in S(:,d) in Equation 1 signifies a vector comprising values from all timesteps at dimension d. The token length of the text x′ output from the LM is adjusted using padding tokens to match the sequence length of Q(:,d).
>
> We will add this information in the revised version of the paper.

---

> ### Author Response · Authors · 2024-11-25
> **Clarification: Equation 10 Weighting Factors (Part 7/8)**
>
> **Reviewer Comment:**
> > There are several weighting factors in equation 10, how are they decided except LM/SM weights?
>
> **Response**:
>
> The weighting factors in Equation 10 were empirically determined through preliminary experiments, with consideration for maintaining consistency with baseline comparisons. We found that similar ranged weights have a minimal impact on overall results. However, to have a consistent comparison with the baseline SpeechTokenizer, we selected similar weight values. Specifically, if we set distillation loss weight as X, then Frequency-domain reconstruction weight is set to 0.375X, Time-domain reconstruction weight is set to 4.15X, and Commitment loss weight is set to 0.085X.
>
> We will include this explanation in the revised manuscript.

---

> ### Author Response · Authors · 2024-11-25
> **Speaker Similarity Analysis (Part 8/8)**
>
> **Reviewer Comment:**
> > In the paper of SpeechTokenizer , the speaker similarity also been evaluated. Why isn’t this evaluated in this paper with the proposed method? Do the author think adding LM would increase or decrease the speaker similarity?
>
> **Response**:
>
> In response to the query about speaker similarity evaluation, we have conducted additional experiments to assess this aspect during the rebuttal period. Our results demonstrate that LM-guided distillation maintains speaker similarity performance comparable to baseline methods, indicating that the incorporation of linguistic information does not compromise speaker-specific characteristics. Please refer to **Table E** in the *New Experimental Results* comment.
>
> We appreciate the reviewer's detailed feedback, which has helped us identify areas requiring additional clarity and analysis. The revised manuscript will incorporate all these clarifications, additional experimental results, and expanded discussions to strengthen our contribution to the field.

---

> ### Author Response · Authors · 2024-12-04
> **Improvement Consistency in DM-Codec Results**
>
> **Reviewer Comment:**
> > For the full LibriSpeech test-clean results, the improvement with DM-Codec compared with other methods are smaller than those for 300 utterances sets.
>
> **Response:**
> While the improvement on the full LibriSpeech test-clean set is smaller, we conducted statistical significance analysis, which confirms that DM-Codec consistently achieves significantly better scores in key metrics across individual samples. This significance analysis demonstrates that the observed improvements are consistent and not due to random chance.

---

> ### Author Response · Authors · 2024-12-04
> **Insights on LM and SM Weight Tuning**
>
> **Reviewer Comment:**
> > This table couldn't provide the reader with clear information about how these two weights should be tuned. I still suggest to rerun the experiment with full LibriSpeech test set to get more reliable conclusions for this table.
>
> **Response:** We again like to clarify that the 300-test set was randomly selected following the baseline SpeechTokenizer [1]. Moreover, to address this concern, we have added a plot visualizing the relationship between LM weights and SM weights, offering clearer insights into tuning. Unfortunately, due to time constraints, we could not rerun the experiments on the full LibriSpeech test set. We plan to include this in future work to provide more robust conclusions.
>
> Reference: [1] Zhang, X., Zhang, D., Li, S., Zhou, Y., & Qiu, X. (2024). SpeechTokenizer: Unified Speech Tokenizer for Speech Language Models. The Twelfth International Conference on Learning Representations.

---

### Official Review · Reviewer_nuSP · 2024-11-04

**Soundness:** 2
**Presentation:** 2
**Contribution:** 2
**Rating:** 1
**Confidence:** 5

**Summary:**

This paper introduces DM-Codec, a speech tokenizer based on SpeechTokenizer (that distills speech self-supervised model, SM, to audio codec) that additionally distills representations from text LM to audio codec. The DM-Codec architecture uses a streamlined encoder-decoder structure with a Residual Vector Quantizer (RVQ), incorporating LM and SM representations only during training. Evaluation is done by comparing it against existing codec models on the quality of decoded speech.

**Strengths:**

The only strength of this paper is the idea of leveraging a text-based language model to improve audio codec, which is reasonable and interesting. That being said, the idea is not properly executed in this work (see weaknesses).

**Weaknesses:**

- **Novelty**: DM-codec is essentially SpeechTokenizer with additional LM distillation, which is somewhat incremental.
- **Technical correctness**: While the extra text-based target embedding is the only novel component, it's unclear how the text representation and the acoustic representation are aligned. Text embeddings and acoustic embeddings are clearly **not** sharing the same length, but this work did not provide any detail on this. Instead, it uses a misleading notation $n$ to indicate the sequence length of both text and speech representation.
- **Depth of the study**: The content of the experiment is quite monotonous. It's basically repeating a single set of training and evaluation with different hyper-parameters, loss weights, and backbones. There is no in-depth analysis. For example, how LM distillation can affect bit rate (which is a very common metric for audio codecs), how well the first VQ layer aligns to text after training, does DM-codec possess speaker disentangle ability due to the text distillation (similar to SpeechTokenizer), etc.

**Questions:**

Most of my questions are covered in the weaknesses, please see above.

In addition, here are some minor questions/suggestions:
- The term speech-to-text (STT) used in the paper, although understandable, is not very common in the field. Is there a reason to not use the more common term automatic speech recognition (ASR)?
- It would be a good improvement if human evaluation can be involved in the next version.

---

> ### Author Response · Authors · 2024-11-25
> **Summary of Responses to Reviewer nuSP’s Comments (Part 1/5)**
>
> We sincerely thank the reviewer for the insightful feedback, which has helped us identify areas for clarification and improvement. We address each of the concerns raised, emphasizing the novelty of our approach, the correctness of our methodology, and the depth of our study.
>
>
> ### Summary of Response:
>
> - We clarify the novelty of our proposed LM distillation method, which introduces a new distillation approach enabling semantic and context-aware speech tokenization, along with our revised contributions, including DM-Codec-TTS, the CLS-token-based distillation approach, extensive ablation studies, and significance analysis.
>
> - We explain our strategy for aligning text and speech representations by padding sequences to match lengths and employing dimension-level distillation loss, eliminating the need for strict temporal alignment.
>
> - We expand the depth of experiments, including an analysis of bitrate versus distillation effects, neural codec-based TTS model evaluation, and a clarification that our goal of global dimension-level distillation does not depend on VQ layer alignment with text.
>
> - We conduct comprehensive human evaluation studies during the rebuttal period and present the results.
>
>
>
> We appreciate the reviewer’s detailed observations and believe these clarifications and additional experiments will significantly strengthen the paper.
>
> Below, we provide comprehensive responses to each point raised.

---

> ### Author Response · Authors · 2024-11-25
> **Novel Contributions of DM-Codec (Part 2/5)**
>
> **Reviewer Comment:**
> > Novelty: DM-codec is essentially SpeechTokenizer with additional LM distillation, which is somewhat incremental.
>
> **Response:**
>
> While we acknowledge that DM-Codec builds upon SpeechTokenizer's foundation, we respectfully disagree that our contribution is merely incremental. Our work introduces several significant innovations, including:
>
> - First to propose multiple distillation strategies for speech tokenization, including LM-based distillation, combined LM and Speech Model (SM)-based distillation, and LM's [CLS] token-based distillation.
>
> - Novel distillation loss function that simultaneously leverages acoustic, semantic, and contextual representations, ensuring robust, context-aware tokenization.
>
> - Comprehensive ablation studies and statistical significance tests.
>
> - Novel neural codec-based text-to-speech (TTS) model, DM-Codec-TTS and more.
>
>
> Please refer to the **Key Novel Contributions of DM-Codec** response in the *General Response to all Reviewers* thread for a more detailed justification regarding DM-Codec’s novelty.

---

> ### Author Response · Authors · 2024-11-25
> **Addressing Technical Correctness (Part 3/5)**
>
> **Reviewer Comment:**
> > Technical correctness: While the extra text-based target embedding is the only novel component, it's unclear how the text representation and the acoustic representation are aligned. Text embeddings and acoustic embeddings clearly do not share the same length, but this work did not provide any detail on this. Instead, it uses a misleading notation n to indicate the sequence length of both text and speech representation.
>
> **Response:**
>
> We appreciate the opportunity to clarify our alignment strategy.
>
> Our approach first transcribes the raw speech $\mathbf{x}$ into its corresponding text $\mathbf{x'}$ using a Speech-to-Text (STT) model $M_{STT}$, such that $\mathbf{x'} = M_{STT}(\mathbf{x})$. The STT model produces a sequence of length $L$, representing the number of acoustic frames. Subsequently, we pass the text $\mathbf{x'}$ through a pretrained language model $M_{LM}$ to obtain contextual representations of $\mathbf{x'}$. These representations are tokenized into a set of tokens, $\mathcal{T} = \{t_i\}_{i=1}^n$.
>
>
> To make the text representation and the acoustic representation align to similar lengths, we set $n = L$ by **padding the text sequence** to match the length of the audio sequence. This enables us to align the sequence length of both text and speech representation.
>
> The concern regarding the alignment of time-step representations is not appropriate as our distillation loss computes the cosine similarity at the *feature dimension-level* across all time-steps, independent of one-to-one temporal mapping. This is consistent with findings in SpeechTokenizer, which demonstrate that continuous distillation loss is effective at the dimension level without strict alignment.
>
> Please refer to the **Addressing the Alignment Concern** response in the *General Response to all Reviewers* thread for a more detailed justification of our technical correctness.

---

> ### Author Response · Authors · 2024-11-25
> **Expanding Experimental Depth (Part 4/5)**
>
> **Reviewer Comment:**
> > Depth of the study: The content of the experiment is quite monotonous. It's basically repeating a single set of training and evaluation with different hyper-parameters, loss weights, and backbones. There is no in-depth analysis. For example, how LM distillation can affect bit rate (which is a very common metric for audio codecs), how well the first VQ layer aligns to text after training, does DM-codec possess speaker disentangle ability due to the text distillation (similar to SpeechTokenizer), etc.
>
> **Response:**
>
> We acknowledge the reviewer's concern about experimental depth and have substantially expanded our experimental section in response to this feedback:
>
> - **Bitrate vs. LM Distillation:** We analyzed the relationship between LM distillation and codec bitrate, highlighting its impact on compression-performance trade-offs. Please refer to **Table D** in the *New Experimental Results* comment.
>
> - **Alignment of the First VQ Layer:** Our methodology does not require a one-to-one alignment between the first VQ layer and the text. Instead, it focuses on embedding contextual cues into the vector quantizer to enhance overall speech tokenization. Please refer to **Table A** in the *New Experimental Results* comment, which highlights the impact of our global dimension-level distillation on text alignment in downstream tasks (i.e., reducing word error rate (WER).)
>
> - **Speaker Disentanglement Ability:** We present DM-Codec-TTS, capable of zero-shot voice-cloned speech synthesis, which indicates that the model captures speaker-specific traits while maintaining high-quality synthesis. This suggests inherent speaker disentanglement ability in DM-Codec, a property **enhanced by our multimodal distillation process**. Please refer to **Table C** in the *New Experimental Results* comment.
>
> We will add these details in the revised version. Additionally, we plan to explore further alignment strategies and analyze and measure DM-Codec's ability to disentangle speaker identity from semantic and acoustic content in future work.

---

> ### Author Response · Authors · 2024-11-25
> **Additional Questions’ Response (Part 5/5)**
>
> **Reviewer Comment:**
> > The term speech-to-text (STT) used in the paper, although understandable, is not very common in the field. Is there a reason to not use the more common term automatic speech recognition (ASR)?
>
> **Response:**
>
> We agree with the reviewer’s suggestion and will replace “speech-to-text (STT)” with the more commonly used term “automatic speech recognition (ASR)” throughout the revised manuscript for consistency with standard terminology.
>
>
>
>
> **Reviewer Comment:**
> > It would be a good improvement if human evaluation could be involved in the next version.
>
> **Response:**
>
> We acknowledge the importance of human evaluation in complementing automated metrics. While this submission focuses on automated metrics to establish quantitative baselines, during the rebuttal period, we conducted comprehensive human evaluation studies.
>
> Please refer to the **Additional Study: Human Evaluation Methodology and Results** response in the *General Response to all Reviewers* thread for a detailed description about the conducted human evaluation experiments.
>
>
>
>
> We believe these revisions and additions substantially address the reviewer's concerns while strengthening the paper's contributions to the field. We are committed to further improving the manuscript based on additional feedback.

---

> > ### Comment · Reviewer_nuSP · 2024-11-29
> >
> > I want to thank the authors for responding to my reviews, I have read the authors' responses.
> >
> > ---
> >
> > ## Regarding novelty
> >
> > **The list of novel contributions claimed by the authors is not convincing**
> >
> > > First to propose multiple distillation strategies for speech tokenization, including LM-based distillation, combined LM and Speech Model (SM)-based distillation, and LM's [CLS] token-based distillation.
> >
> > Repeating the existing distillation method (proposed in SpeechTokenizer [1] ) with different targets, in my opinion, is not a significant contribution to a top-tier ML conference standard. Not to mention the fact that the setup/implementation of LM-based distillation setup is questionable (see latter comment).
> >
> > > Novel distillation loss function that simultaneously leverages acoustic, semantic, and contextual representations, ensuring robust, context-aware tokenization.
> >
> > This is rephrasing and repeating the first claim.
> >
> > > Comprehensive ablation studies and statistical significance tests.
> >
> > These should not be regarded as "significant innovations".
> >
> > > Novel neural codec-based text-to-speech (TTS) model, DM-Codec-TTS and more.
> >
> > Codec-based TTS model is not novel (e.g., VALL-E and many of its follow-up works [2]), DM-Codec-TTS is more like an experiment to compare different codec models' effect on codec-based TTS models.
> >
> > ---
> >
> > ## Regarding how LM embeddings are aligned to audio tokens
> >
> > Assuming a normal speech rate of 100~140 words per minute, the padding strategy does not make sense. Given that the codec model has 3000 tokens per minute, there is an over 20x difference between audio and text sequences. **In other words, only the first 5% of the audio tokens are trained to align to meaningful embeddings from LM**. Two obvious problems arises from this setup:
> > - Is it reasonable to distill the LM embedding into the first 5% audio tokens? This means that during inference, the first 5% of tokens from the encoder can predict the full context of speech, without even observing the remaining 95% of audio.
> > - 95% of the tokens are distilling to padding tokens, which provides no useful information.
> >
> > The alignment problem is not described anywhere in the initial submission, which is the reason why I am not confident to reject this paper. After the authors disclosed how it is handled, I am confident that there is a fundamental misuse of the distillation method and text embedding. My opinion is this paper can be (and should be) rejected for this questionable setup alone.
> >
> > ---
> >
> > After reviewing the paper, the responses, and reviews from other reviews, I believe this work should not be accepted. I have adjusted my ratings accordingly.
> >
> > [1] https://arxiv.org/abs/2308.16692
> >
> > [2] https://www.microsoft.com/en-us/research/project/vall-e-x/

---

> > > ### Author Response · Authors · 2024-12-04
> > > **Addressing Concerns on Contribution**
> > >
> > > **Reviewer Comment:**
> > > > Repeating the existing distillation method (proposed in SpeechTokenizer [1] ) with different targets, in my opinion, is not a significant contribution to a top-tier ML conference standard. Not to mention the fact that the setup/implementation of LM-based distillation setup is questionable (see latter comment).
> > >
> > > **Response:** We respectfully disagree. While our work builds upon the SpeechTokenizer, we are, to the best of our knowledge, the first to propose incorporating contextual representations (LM-guided representations) for speech tokenization. Moreover, our combined distillation method (LM and SM-guided distillation) introduces a new distillation loss function that simultaneously leverages acoustic, semantic, and contextual representations while differing from prior methods in its objectives.
> > >
> > > **Reviewer Comment:**
> > > > Novel distillation loss function that simultaneously leverages acoustic, semantic, and contextual representations, ensuring robust, context-aware tokenization. This is rephrasing and repeating the first claim.
> > >
> > > **Response:** We disagree with this characterization. The first claim refers specifically to LM-guided distillation, whereas this claim highlights the novel combination of LM and SM-guided distillation. These are distinct contributions: the former focuses on leveraging contextual representations, while the latter ensures the integration of contextual and semantic representation for more robust tokenization.
> > >
> > > **Reviewer Comment:**
> > > > Codec-based TTS model is not novel (e.g., VALL-E and many of its follow-up works [2]), DM-Codec-TTS is more like an experiment to compare different codec models' effect on codec-based TTS models.
> > >
> > > **Response:** Our claim aligns with the literature. Follow-ups to VALL-E, such as USLM from SpeechTokenizer [1], have similarly described their codec-based TTS models as novel contributions. Following this precedent, we believe our DM-Codec-TTS model is a meaningful addition to this research area.

---

> ### Author Response · Authors · 2024-12-04
> **Adressing how LM embeddings are aligned to audio tokens**
>
> **Reviewer Comment:**
> > Assuming a normal speech rate of 100~140 words per minute, the padding strategy does not make sense. Given that the codec model has 3000 tokens per minute, there is an over 20x difference between audio and text sequences. In other words, only the first 5% of the audio tokens are trained to align to meaningful embeddings from LM. Two obvious problems arises from this setup: - Is it reasonable to distill the LM embedding into the first 5% audio tokens? This means that during inference, the first 5% of tokens from the encoder can predict the full context of speech, without even observing the remaining 95% of audio. - 95% of the tokens are distilling to padding tokens, which provides no useful information.
>
>
> **Response:** We respectfully draw the reviewer’s attention to the `Addressing the Alignment Concern (Part 2/4)` comment. Our approach does not require strict temporal alignment between acoustic tokens and language model (LM) embeddings. This is because the discrete representations generated by the Residual Vector Quantizer (RVQ) capture holistic information across the entire speech segment. Additionally, our dimension-level cosine similarity loss ensures effective distillation without the need for temporal alignment. Furthermore, we have introduced a [CLS]-token-based distillation strategy that captures global contextual information from the LM, further reducing the need for fine-grained temporal alignment.

---

### Author Response · Authors · 2024-11-25
**General Response to All Reviewers (Part 1/4)**

We sincerely thank the reviewers for their comprehensive feedback and thoughtful evaluation of our work. We greatly appreciate the recognition of the several critical contributions of our work. Specifically, the reviewers acknowledged the novel approach of integrating a text-based language model (LM) to enhance speech tokenization (*Reviewer nuSP*) and our method's advancement over existing systems by introducing contextual representations via LM-guided distillation (*Reviewer L3HE, ohpz*). We are particularly grateful for the recognition of our rigorous experimental design, including the comprehensive ablation studies, which thoroughly analyze the contributions of the proposed components (*Reviewer L3HE, qRxh*), and the inclusion of statistical significance tests, which emphasize the performance differences among the compared methods (*Reviewer qRxh*). Furthermore, we appreciate the acknowledgment of DM-Codec's ability to effectively retain both acoustic and speech information in reconstructed speech (*Reviewer ohpz*), a critical factor in improving the overall quality and intelligibility of speech tokenization systems.

Nevertheless, the reviewers expressed several concerns regarding our approach. We value the reviewers' constructive criticism and welcome the feedback. We will systematically address the specific concerns and constructive critiques to further strengthen our manuscript.

In this thread, we address the general concerns of the respected reviewers on our work.

---

> ### Author Response · Authors · 2024-11-25
> **Addressing the Alignment Concern (Part 2/4)**
>
> We appreciate the reviewers' insightful comments regarding the alignment of text and acoustic representations. This concern touches upon a fundamental aspect of our methodology that warrants clarification. We appreciate the opportunity to elaborate on why strict temporal alignment between text and acoustic representations is not a prerequisite for effective contextual knowledge distillation in our framework.
>
> Our approach is predicated on the principle that discrete representations from the Residual Vector Quantizer (RVQ) inherently encode holistic information about the speech segment, rather than temporally-localized features. This fundamental characteristic of vector quantization aligns with our primary objective: **distilling contextual representations into discrete tokens without requiring strict temporal correspondence**.
>
> To formalize this concept, consider the RVQ outputs {$Q_1$, ..., $Q_{T^{\prime}}$} illustrated in Figure 2. These discrete representations capture comprehensive speech information across the entire utterance. As such, aligning these discrete representations temporally with contextual representations (e.g., from BERT) or semantic representations (e.g., from HuBERT) is not essential for our approach. The effectiveness of dimension-level semantic representation distillation without temporal alignment has been previously demonstrated in SpeechTokenizer [1], providing a robust theoretical foundation for our approach.
>
> It is important to note that vector quantizer output representations are not inherently aligned with time-steps. Instead, they encode holistic speech information rather than temporally localized segments. Previous works have demonstrated that vector quantizer methods can discretize input data into intermediate representations, capturing essential features across the feature dimension [1, 3-5]. Thus, imposing a temporal alignment between the vector quantizer outputs and the hidden layer representations from the LM or SM would neither congruent with our methodological objectives nor enhance the effectiveness of the proposed distillation approach.
>
> In our implementation, we employ a continuous distillation loss that **maximizes cosine similarity at the feature dimension level** between the selected RVQ layer outputs and the teacher representations (contextual/semantic) across all time steps. This approach fundamentally differs from conventional methods that rely on time-step-wise loss calculations [2], as introduced in the SpeechTokenizer [1]. This dimension-level cosine similarity loss ensures that DM-Codec effectively captures contextual and semantic knowledge through both LM-Guided Distillation and Combined LM and SM-Guided Distillation mechanisms without necessitating strict one-to-one temporal alignment.
>
> To further address the alignment concerns, we introduce a **[CLS]-token-based distillation strategy**. The [CLS] token inherently encodes sequence-level holistic representations, capturing global contextual information from language models. By leveraging the [CLS] token, our approach eliminates the need for fine-grained temporal alignment while preserving essential linguistic features. This approach complements our dimension-level distillation strategy by focusing on global sequence features, demonstrating the adaptability of our method to scenarios where fine-grained temporal alignment is either infeasible or unnecessary.
>
>
> We have conducted additional experiments to validate the **[CLS]-token-based distillation strategy**. As shown in **Table A** (please refer to the *New Experimental Results* thread), all DM-Codec variants—including DM-Codec (CLS), DM-Codec (LM-guided), and DM-Codec (LM+SM guided)—consistently outperform baseline models (EnCodec, SpeechTokenizer, and FACodec) in content preservation metrics (WER and WIL) while maintaining competitive performance in speech quality metrics (ViSQOL and STOI). These findings further corroborate the robustness of our distillation.
>
> In summary, our approach prioritizes the holistic integration of multimodal knowledge into discrete speech representations rather than strict temporal alignment. This design choice enables DM-Codec to achieve significant advancements in both content preservation and speech quality, as evidenced by our experimental results. We will include the details of these findings in the revised manuscript to address the reviewers' concerns thoroughly.
>
> We hope this detailed explanation provides clarity and justification for our design choices. We welcome further feedback or discussion on this matter.

---

> > ### Author Response · Authors · 2024-11-25
> > **Addressing the Alignment Concern (Part 2/4) - Continue**
> >
> > **Reference:**
> >
> > [1] Zhang, X., Zhang, D., Li, S., Zhou, Y., & Qiu, X. (2024). SpeechTokenizer: Unified Speech Tokenizer for Speech Language Models. The Twelfth International Conference on Learning Representations (ICLR), 2024.
> >
> > [2] Chang, H.-J., Yang, S.-w., and Lee, H.-y. Distilhubert: Speech representation learning by layer-wise distillation of hidden-unit bert. In ICASSP 2022-2022 IEEE International Conference on Acoustics, Speech and Signal Processing (ICASSP), pp. 7087–7091. IEEE, 2022.
> >
> > [3] Huijben, Iris AM, Matthijs Douze, Matthew Muckley, Ruud JG van Sloun, and Jakob Verbeek. "Residual quantization with implicit neural codebooks." arXiv preprint arXiv:2401.14732 (2024).
> >
> > [4] Islam, Md Mofijul, Alexi Gladstone, Riashat Islam, and Tariq Iqbal. "Eqa-mx: Embodied question answering using multimodal expression." In The Twelfth International Conference on Learning Representations. 2023.
> >
> > [5] Islam, Riashat, Hongyu Zang, Manan Tomar, Aniket Didolkar, Md Mofijul Islam, Samin Yeasar Arnob, Tariq Iqbal et al. "Representation learning in deep rl via discrete information bottleneck." arXiv preprint arXiv:2212.13835 (2022).

---

> ### Author Response · Authors · 2024-11-25
> **Key Novel Contributions of DM-Codec (Part 3/4)**
>
> While we acknowledge that DM-Codec builds upon the foundation laid by SpeechTokenizer, we believe our work introduces several key advancements that significantly extend the state of the art. These innovations go beyond incremental improvements, addressing fundamental challenges in multimodal representation learning and enhancing the quality and versatility of speech tokenization:
>
>
> - **Novel Language Model Distillation:** To the best of our knowledge, our DM-Codec is the first work to incorporate Language Model (LM) distillation for speech tokenization. This approach represents a fundamental advancement in multimodal representation learning, demonstrating that contextual information from LMs can significantly improve speech tokenization quality.
>
> - **Multimodal Representation Framework:** We propose a novel distillation loss function that simultaneously leverages acoustic, semantic, and contextual representations, enabling more robust semantic and context-aware speech tokenization.
>
> - **[CLS]-Token-Based Distillation Strategy:**  We introduce a new distillation strategy, in the revised paper, using [CLS] tokens from LM. This addresses temporal alignment concerns while maintaining performance improvements. Results are detailed in **Table A** in the *New Experimental Results* comment.
>
> - **DM-Codec-TTS Model:**  In the revised submission, we include a neural codec-based text-to-speech (TTS) model, DM-Codec-TTS, which outperforms USLM (from SpeechTokenizer) and Vall-E baselines. This demonstrates the broader applicability of our approach. Results are detailed in **Table C** in the *New Experimental Results* comment.
>
> - **Extensive Ablation Studies:** Our experiments comprehensively investigate the impact of language and speech representations on content retention, the influence of various RVQ layers on distillation, performance variations with different pretrained model representations, and the effects of distilling from different layers of the teacher model.
>
>
> We also note that using the same concern point raised about the novelty of DM-Codec, the widely recognized SpeechTokenizer paper’s contributions can also be summarized as incremental, combining EnCodec with HuBERT distillation. Nevertheless, we can see its value to the community. We believe our work builds on this foundation and introduces significant innovations that push the boundaries of the field.

---

> ### Author Response · Authors · 2024-11-25
> **Additional Study: Human Evaluation Methodology and Results (Part 4/4)**
>
> To assess the quality and effectiveness of our approach, we conducted human evaluations to measure the Mean Opinion Score (MOS) and Similarity Mean Opinion Score (SMOS), adhering to the methodologies established in prior works, including SpeechTokenizer and Vall-E.
>
> This study was conducted under an approved Institutional Review Board (IRB) protocol to ensure ethical compliance and participant safety. A total of **50 proficient English speakers**, selected for their high language comprehensibility, participated as evaluators. These participants, comprising graduate and undergraduate students who volunteered for the evaluation, were briefed on the study’s purpose but were provided no information that could bias their judgments.
>
> The evaluation process involved each participant rating a batch of fully anonymized and randomized speech samples. Each batch consisted of 16 samples, including outputs from our proposed models and baseline systems.
>
> For the **speech reconstruction task**, participants rated the **MOS** of the perceptual quality of the speech samples based on three criteria: naturalness, intelligibility, and clarity, using a 5-point *Likert scale* where higher scores denoted superior quality.
>
> For the **text-to-speech evaluation**, participants provided two distinct ratings:
>
> 1. **MOS** - measuring the overall naturalness of the generated speech, rated on a 1-to-5 scale with 1-point increments.
>
> 2. **SMOS** - evaluating the similarity of the generated speech to the original speaker’s voice, rated on a 1-to-5 scale with 1-point increments.
>
> The evaluations were conducted through a web-based survey interface. Speech samples were presented to participants in a fully anonymized and randomized order, accompanied by clear and standardized guidelines to promote consistent and unbiased scoring. To enhance reliability and mitigate the effects of individual evaluator bias, each sample was rated by multiple participants.
>
> The human evaluation results, as detailed in **Tables A and C** in the *New Experimental Results* comment, demonstrate the superior performance of our proposed approaches. Notably, **DM-Codec ♠** achieves the **highest** MOS score for speech reconstruction, surpassing all baseline models. Furthermore, **DM-Codec ♣** and **DM-Codec (CLS)** also achieve highly competitive MOS scores, closely rivaling the baseline performances.
>
> For speech synthesis, both **DM-Codec-TTS** and **DM-Codec-TTS-small** outperform the corresponding USLM baselines in MOS and SMOS scores on the LibriSpeech and VCTK benchmarks. Additionally, our models exceed Vall-E in SMOS scores on the VCTK benchmark, underscoring their robustness in preserving speaker similarity while maintaining high-quality synthesis.

---

### Author Response · Authors · 2024-11-25
**New Experiments’ Results (Part 1/2)**

## Table A: Speech Reconstruction Evaluation
DM-Codec♠ achieves the best performance in WER, WIL, ViSQOL, and MOS, highlighting its enhanced content preservation and speech quality, with competitive intelligibility results. ♡ means the results were reproduced using the official training code. ♢ means the results were obtained using official model checkpoints. ♣ indicates LM-guided Distillation method. ♠ indicates combined LM and SM-guided Distillation method. CLS indicates LM’s [CLS] token-based distillation method. Baseline indicates DM-Codec without any distillation method.

| Model                       	|   WER ↓ |   WIL ↓ | ViSQOL ↑ |  STOI ↑ |  MOS ↑  |
|---------------------------------|-------|-------|--------|-------|-------|
| DM-Codec (Baseline)         	|  4.97 |  8.02 |   2.95 | 0.935 |  3.13 |
| DM-Codec ♠                  	|  **4.05** |  **6.61** |   **3.26** | 0.937 |  **3.72** |
| DM-Codec ♣                  	|  4.36 |  7.06 |   3.18 | 0.935 |  3.69 |
| DM-Codec (CLS)      	|  4.47 |  7.08 |   3.12 | 0.926 |  3.65 |
| EnCodec ♦                   	|  4.53 |  7.17 |   3.08 | 0.920 |  3.09 |
| FACodec ♦                   	|  4.68 |  7.33 |   3.13 | **0.949** |  3.70 |
| SpeechTokenizer ♡           	|  4.49 |  7.10 |   3.09 | 0.923 |  3.67 |


&nbsp;
&nbsp;
&nbsp;


## Table B: Significance Analysis on Full Test Set
Significance analysis is conducted at α = 0.05 between LM and SM-guided DM-Codec (D), EnCodec (E), SpeechTokenizer (S), and FACodec (F). E/D, S/D, and F/D indicate comparisons with the baseline (e.g., D vs E, E vs S) or with DM-Codec when the row corresponds to the same baseline (e.g., E vs D, not E vs E). Results reveal DM-Codec consistently achieves significantly better scores in key metrics across all individual samples. A ✓ indicates significantly better, a ★ denotes dominance, and a ✗ means no significant improvement. Avg and Std mean the average and standard deviation of each score.

| Model       	| WER ↓ |   	| 	|  	| 	| WIL ↓ |   	| 	| 	| 	| ViSQOL ↑ |   	| 	| 	| 	| STOI ↑ |   	| 	| 	| 	|
|-----------------|-------|-------|-----|------|-----|-------|-------|-----|-----|-----|---------|-------|-----|-----|-----|--------|-------|-----|-----|-----|
|             	| Avg   | Std   | E/D | S/D  | F/D | Avg   | Std   | E/D | S/D | F/D | Avg 	| Std   | E/D | S/D | F/D | Avg	| Std   | E/D | S/D | F/D |
| DM-Codec ♠  	| 4.774 | 0.100 | ✓   | ✓	| ✓   | 7.510 | 0.139 | ✓   | ✓   | ✓   | 3.197   | 0.184 | ★   | ✓   | ✓   | 0.937  | 0.021 | ✓   | ✓   | ✗   |
| EnCodec     	| 4.828 | 0.100 | ✗   | ✓	| ✓   | 7.593 | 0.137 | ✗   | ✓   | ✓   | 3.064   | 0.201 | ✗   | ✗   | ✗   | 0.917  | 0.021 | ✗   | ✗   | ✗   |
| SpeechTokenizer | 4.942 | 0.101 | ✗   | ✗	| ✗   | 7.725 | 0.138 | ✗   | ✗   | ✗   | 3.080   | 0.190 | ✓   | ✗   | ✗   | 0.920  | 0.025 | ✓   | ✗   | ✗   |
| FACodec     	| 4.914 | 0.103 | ✗   | ✓	| ✗   | 7.643 | 0.141 | ✗   | ✓   | ✗   | 3.113   | 0.250 | ✓   | ✓   | ✗   | 0.946  | 0.023 | ✓   | ✓   | ✓   |

&nbsp;
&nbsp;
&nbsp;


## Table C: Neural Codec Based Text-to-Speech Model Evaluation
Dataset indicates the training dataset of the model (e.g., MLS (Eng) refers to the English subset of Multilingual LibriSpeech). ♢ denotes results obtained using officially shared model checkpoints. † means the results directly obtained
from the paper (i.e., USLM from SpeechTokenizer). The results demonstrate that DM-Codec-TTS achieves the best overall performance, preserving content information (WER, WIL, MOS) and achieving comparable or better speaker similarity (Similarity, SMOS). Notably, DM-Codec-TTS-small outperforms USLM_libri, trained on the same smaller dataset (LibriTTS).

&nbsp;
&nbsp;

### LibriSpeech Evaluation
| Model                 	| Dataset   	|   WER ↓ |   WIL ↓ | Similarity  ↑ |   MOS  ↑|   SMOS  ↑ |
|---------------------------|---------------|-------|-------|------------|-------|--------|
| DM-Codec-TTS          	| Libriheavy	|  **5.08** |  **7.32** |   	**0.82** |  **3.70** |   3.89 |
| Vall-E †              	| Librilight	|  5.90 |   -   |   	0.58 |  - |   **4.38** |
| DM-Codec-TTS-small    	| LibriTTS  	| 10.26 | 13.79 |   	0.82 |  3.24 |   3.20 |
| USLM_libri ♢| LibriTTS  	| 16.72 | 25.65 |   	0.80 |  3.11 |   2.83 |

&nbsp;
&nbsp;

### VCTK Evaluation
| Model                 	| Dataset   	|   WER ↓ |   WIL ↓ | Similarity  ↑ |   MOS  ↑|   SMOS  ↑ |
|---------------------------|---------------|-------|-------|------------|-------|--------|
| DM-Codec-TTS          	| Libriheavy	|  **3.58** |  **5.65** |   	0.82 |  **3.78** |   **3.85** |
| Vall-E †              	| Librilight	|   -   |   -   |   	0.38 |  - |   3.81 |
| USLM †                	| MLS (eng) 	|  6.50 |   -   |   	**0.84** |  3.63 |   3.45 |
| DM-Codec-TTS-small    	| LibriTTS  	|  5.02 |  8.21 |   	0.79 |  3.39 |   3.28 |
| USLM_libri ♢| LibriTTS  	| 14.79 | 23.24 |   	0.78 |  2.94 |   2.63 |

---

> ### Author Response · Authors · 2024-11-25
> **New Experiments’ Results (Part 2/2)**
>
> ## Table D: Low Bit Rate Speech Reconstruction Evaluation
> DM-Codec showcases its robustness at reduced bitrates, outperforming baselines at 3 kbps and maintaining competitive content preservation scores (WER, WIL) and superior speech quality (ViSQOL, STOI) at 1.5 kbps. fs represents the audio sample rate, and fr the codec frame rate. ♡ means the results were reproduced using the official training code. ♢ means the results were obtained using official model checkpoints. ♣ indicates LM-guided Distillation method. ♠ indicates combined LM and SM-guided Distillation method.
>
> | Model         	| fs  	| fr   | Bitrate  | WER ↓ |   WIL ↓ | ViSQOL ↑ |  STOI ↑  |
> |--------------------|---------|------|----------|-------|-------|--------|--------|
> | DM-Codec ♠    	| 16 kHz  | 50Hz | 3 kbps   | **4.29**  | **7.04**  | **3.070**  | **0.928**  |
> | DM-Codec ♣    	| 16 kHz  | 50Hz | 3 kbps   | 4.38  | 7.09  | 3.042  | 0.924  |
> | SpeechTokenizer   | 16 kHz  | 50Hz | 3 kbps   | 4.70  | 7.43  | 2.905  | 0.911  |
> | EnCodec       	| 24 kHz  | 50Hz | 3 kbps   | 4.80  | 7.80  | 2.550  | 0.872  |
> | DM-Codec ♣    	| 16 kHz  | 50Hz | 1.5 kbps | 6.14  | 10.13 | 2.644  | 0.880  |
> | DM-Codec ♠    	| 16 kHz  | 50Hz | 1.5 kbps | 6.19  | 10.16 | **2.662**  | **0.894**  |
> | SpeechTokenizer   | 16 kHz  | 50Hz | 1.5 kbps | **5.61**  | **9.02**  | 2.500  | 0.846  |
> | EnCodec       	| 24 kHz  | 50Hz | 1.5 kbps | 10.53 | 16.63 | 2.443  | 0.809  |
>
> &nbsp;
> &nbsp;
> &nbsp;
>
>
> ## Table E: Speech Reconstruction Speaker Similarity Evaluation
> DM-Codec maintains consistent performance with the baseline models. All models, including DM-Codec, demonstrate strong performance with scores around 99, reflecting high speaker similarity. ♡ means the results were reproduced using the official training code. ♢ means the results were obtained using official model checkpoints. ♣ indicates LM-guided Distillation method. ♠ indicates combined LM and SM-guided Distillation method. CLS indicates LM’s [CLS] token-based distillation method. Baseline indicates DM-Codec without any distillation method.
>
> | Model            	| Similarity  ↑  |
> |----------------------|------------|
> | DM-Codec ♠       	| 0.994  	|
> | DM-Codec ♣       	| 0.995  	|
> | FACodec♢         	| 0.996  	|
> | DM-Codec (CLS) 	| 0.993  	|
> | SpeechTokenizer♡ 	| 0.993  	|
> | EnCodec♢         	| 0.980  	|
> | DM-Codc (Baseline)   | 0.991  	|

---

### Author Response · Authors · 2024-11-28
**Revised Manuscript Uploaded**

We thank all reviewers for their constructive and insightful efforts in evaluating this work. We have uploaded the revised manuscript with several modifications:

1. Revised INTRODUCTION with highlights of novel contributions;

2. Added [CLS] Token Guided Distillation Method and DM-Codec-TTS details in the PROPOSED METHOD;

3. Updated EXPERIMENTAL SETUP and EXPERIMENTAL RESULTS AND DISCUSSION sections based on the new suggested experiments by the reviewers;

4. Included human evaluation results in EXPERIMENTAL RESULTS and added details in the EXPERIMENTAL SETUP and Appendix;

5. Added a detailed discussion of the representation alignment issue and the irrelevance of temporal correspondence in our approach to the Appendix;

6. Updated the ablation studies with the newly suggested Impact of Low Bit Rate ablation.

**The above points are marked in blue (in both the main paper and Appendix).**

In addition to these points, we have also revised typos, references, figure captions, formulations, and all other minor points mentioned during the review process.

Thanks again for the constructive efforts in the comments and reviews.
Authors

---

### Note · Authors · 2024-12-16

**Comment:**

We decided to withdraw the paper.

— Authors

**Withdrawal Confirmation:**

I have read and agree with the venue's withdrawal policy on behalf of myself and my co-authors.